



# Quantifying ice crystal growth rates in natural clouds from glaciogenic cloud seeding experiments

Christopher Fuchs[1], Fabiola Ramelli[1], Anna J. Miller[1], Nadja Omanovic[1], Robert Spirig[1], Huiying Zhang[1], Patric Seifert[2], Kevin Ohneiser[2], Ulrike Lohmann[1], and Jan Henneberger[1]

[1]Institute for Atmospheric and Climate Science, ETH Zurich, Zurich, 8092 Switzerland
[2]Leibniz Institute for Tropospheric Research (TROPOS), Leipzig, Germany

**Correspondence:** Christopher Fuchs (christopher.fuchs@env.ethz.ch) and Jan Henneberger (jan.henneberger@env.ethz.ch)

**Abstract.** Ice crystals are essential in the evolution of mixed-phase clouds, as ice crystals can quickly grow to large sizes by vapor diffusion and thereby trigger precipitation formation. Vapor diffusional growth rates of ice crystals were quantitatively studied in the laboratory for several decades, forming the basis of various ice crystal growth models. Since field measurements generally only provide snapshots that lack information on ice crystal age or changes induced by cloud processes, significant

gaps remain in quantitative field observations impeding the validation of laboratory experiments and models.

Our study addresses this gap through innovative glaciogenic cloud seeding experiments in persistent low-level stratus clouds in the CLOUDLAB project. The controllability and repeatability of our seeding experiments facilitates quantifying diffusional ice crystal growth rates in natural clouds via in situ measurements. We report growth rates of 0.17–0.81 $\mu m\,s^{-1}$ (major axis of pristine ice crystals) from 14 seeding experiments between -5.1 to -8.3 °C. We also observe how microphysical characteristics

induce strong variations in the growth rates, e.g., reduced growth rates in seeding-induced regions of high ice crystal number concentrations. For better comparison to laboratory and non-seeded clouds, we developed two filtering methods to isolate growth conditions less affected by the experimental setup. The comparison shows that the temperature-dependent growth rate variations align with laboratory data, whereas absolute laboratory values are higher. Our findings provide valuable insights into the vapor diffusional growth of ice crystals in natural clouds and connect in situ observations with laboratory and modeling

studies.

## 1 Introduction

Ice crystals play a crucial role in the Earth's atmosphere, as they influence the evolution of clouds (Korolev et al., 2017) and are a cornerstone for precipitation formation (Mülmenstädt et al., 2015; Heymsfield et al., 2020). They are particularly important in the context of the lifetimes of mixed-phase clouds (MPCs) (Korolev et al., 2017), which are the main source of precipitation

over most land masses. MPCs can exists at temperatures between 0 °C and -38 °C, consisting of both metastable supercooled cloud droplets and ice crystals, and are therefore in a thermodynamically unstable equilibrium. After the nucleation of ice crystals in MPCs, the ice crystals can grow rapidly to large sizes by vapor diffusion due to the lower vapor pressure over ice than over water (Korolev, 2007; Korolev et al., 2017). In a special case, where the ambient vapor pressure falls between




the vapor pressure over ice and water, ice crystals grow at the expense of evaporating cloud droplets, which is known as the
Wegener–Bergeron–Findeisen (WBF) process (Wegener, 1911; Bergeron, 1935; Findeisen, 1938). This rapid ice growth can
trigger subsequent ice-phase processes such as riming and aggregation, paving the way for precipitation formation and / or full
glaciation of MPCs.

The growth of ice crystals by vapor diffusion has been thoroughly studied in the laboratory over a wide temperature range
for more than 70 years (e.g., Vonnegut, 1947; Mason, 1953; Nakaya, 1954; Kobayashi, 1961; Hallett, 1965; Keller and Hallett,
1982; Lamb and Scott, 1972; Ryan et al., 1974, 1976; Takahashi and Fukuta, 1988; Takahashi et al., 1991; Bailey and Hallett,
2004; Castellano et al., 2014). These studies investigated the temperature and water-saturation-dependent formation of ice
crystal habits (i.e., plates vs. columns) and how fast the ice crystals grow under the respective ambient conditions (i.e, ice
crystal growth rates). These laboratory studies build the foundation of numerous theoretical concepts and models (e.g., Kuroda
and Lacmann, 1982; Cotton, 1972; Lamb and Scott, 1974; Miller and Young, 1979; Chen and Lamb, 1994; Nelson and Baker,
1996; Wood et al., 2001; Hashino and Tripoli, 2008; Zhang and Harrington, 2014; Harrington and Pokrifka, 2021). However,
quantitative in situ field observations of vapor diffusional ice crystal growth rates in natural clouds, which are needed to validate
laboratory and theoretical results, are mostly missing, because field studies are generally limited by a lack of controllability
and repeatability. Therefore, field observations usually provide only a snapshot of the unique microphysical states of a cloud,
and since every cloud is different with various processes occurring simultaneously, it is often impossible to infer the process
rates or to generalize the findings. The studies by Field (1999) and Heymsfield et al. (2011) addressed this issue by using
quasi-Lagrangian measurements on aircraft. They observed the evolution of ice particles in altostratus at temperatures -20 to
-40 °C, and also quantified the linear and mass growth rates of ice crystals in lenticular clouds in the temperature range -32 to
-40 °C, respectively.

To overcome the lack of quantitative field observations, we use glaciogenic cloud seeding under the framework of the
CLOUDLAB project (see Henneberger et al., 2023) to perform confined and controlled experiments inside supercooled low
stratus clouds over the Swiss Plateau. The small scale and high temporal repeatability of the experiments allows us to study
vapor diffusional ice crystal growth rates at different temperatures, as demonstrated by Ramelli et al. (2024), where the authors
gave proof-of-concept to our experimental approach for 4 seeding experiments at two temperatures. Now, we present observa-
tions of 14 seeding experiments, at twelve different temperatures between -5.1 to -8.3 °C, and examine the impacts of varying
microphysical conditions on the vapor diffusional growth of ice crystal and compare them with laboratory studies.

## 2 Data and methods

All analyzed data were collected in the CLOUDLAB project during three field campaigns from January 2022 - March 2022,
December 2022 - February 2023, and December 2023 - February 2024. The CLOUDLAB main site is located in the Swiss
Plateau, near Eriswil (47°04'14"N, 7°52'22"E; 920 m a.s.l.), where a versatile set of remote sensing instruments was installed
(for more information see Henneberger et al. (2023)), along with a tethered balloon system (TBS) for in situ measurements.
The TBS was equipped with the holographic imager for microscopic objects (HOLIMO, see Sect. 2.2 and Ramelli et al., 2020),



which measures phase- and size-resolved cloud microphysical properties by capturing images of hydrometeors. All methods and instruments relevant to this study are described in more detail below.

## 2.1 Glaciogenic cloud seeding experiments

The CLOUDLAB approach uses glaciogenic cloud seeding in supercooled low stratus to study microphysical processes, e.g., the growth rates of ice crystals, in natural clouds. The targeted low stratus clouds are predominantly liquid, with seeding temperatures below -5 °C, and a cloud base below ≈ 1000 m a.g.l. (≈ 2000 m a.s.l.). The course of a seeding experiment is visualized in Fig. 1. A customized uncrewed aerial vehicle (UAV, Meteodrone MM-670, Meteomatics AG, Switzerland, see Miller et al., 2024b) is equipped with a burn-in-place flare (Zeus MK2, Cloud Seeding Technologies, Germany) containing ≈ 200 g of seeding particles (mixture of silver iodide (≈ 20g) and other compounds, which is ice-active at temperatures below -5 °C Chen et al., 2024; Miller et al., 2024a). The UAV releases seeding particles for 5 to 6 minutes (flare burning time) into the cloud at a distance between 1 km and 3 km upwind of the main main site, either stationary or flying legs (200 m - 400 m) perpendicular to the wind direction at constant altitude and flight speed (see Fig. 3 in Miller et al., 2024b). The freshly nucleated ice crystals are then advected towards the main site and grow by vapor diffusion along the way until the seeding plume (mixture of ice crystals and cloud droplets) reaches the main measurement site. There, the seeding plume is observed by in situ measurements with the TBS carrying HOLIMO, and by remote sensing instruments, using up to three cloud radars simultaneously (two 35.12 GHz Ka-band scanning Doppler cloud radars, Mira-35, Metek, Germany, (Görsdorf et al., 2015); one 94 GHz frequency modulated continuous wave W-band vertical pointing radar, RPG-FMCW-94, RPG Radiometer physics GmbH, Germany, (Küchler et al., 2017)). The unseeded cloud is monitored before and after the passage of the seeding plume with the same instrumentation to infer the cloud microphysical properties of the undisturbed background cloud, which also helps to better identify the seeding-induced changes. A more comprehensive description of the seeding experiments and instrumentation can be found in Henneberger et al. (2023).

Since the analysis of holographic data is very time intensive we were able to process total of 20 experiments, which were selected to cover the widest range of the observed seeding temperatures. However, this study includes only 14 of these experiments, because the others did not have a sufficiently high number of pristine ice crystals for the growth rate analysis. The 14 experiments were carried out on six different days, over a temperature range from -5.1 °C to -8.3 °C, with ice crystal residence times, i.e., the time between seeding and observation, ranging from ≈4.8 min to ≈12 min. All experiments and relevant parameters are given in Table A1 and observations for an exemplary seeding experiment are shown in the Sect. 2.3.

## 2.2 In situ observations using a tethered balloon system and a holographic imager

A TBS (see Ramelli et al., 2020) is used to lift our in situ instrumentation inside the cloud. The TBS consists of a $200\,\text{m}^3$ kytoon (a cross between balloon and kite, Allsopp Helikite, United Kingdom) filled with helium. The measurement platform hangs ≈ 30 m below the kytoon and carries different in situ instruments. Depending on the weight of the measurement platform, wind speed, and icing conditions, a maximum height of 1 km a.g.l. can be reached.





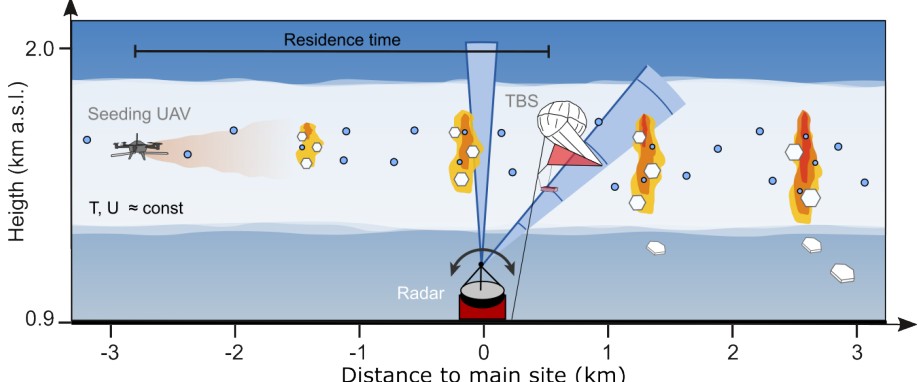

**Figure 1.** Visualization of the CLOUDLAB seeding experiments. An uncrewed aerial vehicle (UAV) releases seeding particles inside a predominantly liquid and supercooled low stratus (white shaded area, blue circles) between 2 to 3 km upwind of the CLOUDLAB main site (located at 0 km). The release of seeding particles initiates the nucleation of ice crystals, which grow via vapor diffusion as they are advected towards the main site. At the main site, the seeding plume is observed by in situ measurements with a tethered balloon system (TBS) carrying a holographic imager (hanging below) and by ground-based remote sensing instruments with up to three vertical-pointing and scanning cloud radars. The size of the ice crystals depends on their residence time inside the cloud, as indicated by the increasing size of the hexagonal plates (from left to right) and the intensification of the radar reflectivity (red-yellow shading). Figure adapted from Ramelli et al. (2024, Fig. 1).

The TBS measurement platform carries HOLIMO, an in situ cloud probe based on digital in-line holography. HOLIMO
measures cloud droplets and ice crystals with diameters > 6 µm and > 25 µm, respectively. A detailed description of the instrument and its operation can be found in Ramelli et al. (2020). The holographic data was processed using the HoloSuite software package (Fugal et al., 2009), and the retrieved hydrometeors were classified into cloud droplets and ice crystals based on the particle shape, using a fine-tuned version of the convolutional neural network introduced by Touloupas et al. (2020). In addition, cloud droplets larger than ≈ 35 µm and all ice crystals were manually visually confirmed and re-labeled if falsely
classified by the neural network. Ice crystals were further manually classified into the categories of pristine and non-pristine (i.e., aggregated and rimed ice crystals). Based on Beck (2017), the uncertainty for cloud droplet number concentration is ≈ ± 5 % and uncertainty for ice crystal number concentration is between 5 % and 10 % for ice crystals larger than 100 µm in diameter and ≈ 15 % for smaller ice crystals.

All experiments were recorded with the maximum sample volume rate of ≈ 0.8 L s$^{-1}$ (sample volume: ≈ 20 ml, maximum
sampling rate: 40 Hz). Due to the high computational effort required to process the holographic data, seeding experiments 1, 2, 10, and 11 (see Table A1) were analyzed with a frequency of 20 Hz (part of the study by Ramelli et al. (2024)) and all other seeding experiments with a frequency of 5 Hz. The background cloud conditions before and after the passage of the seeding plume were analyzed with a frequency of 10 Hz and 1 Hz, respectively.





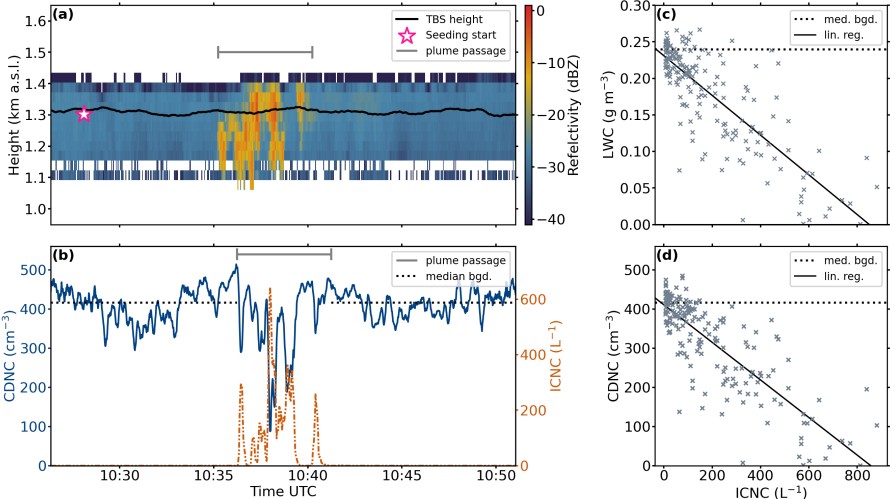

**Figure 2.** Observations for seeding experiment 3, conducted on 25.01.2023 at -5.5 °C. **(a)**: A height-time indicator (HTI) radar measurement shows the undisturbed background cloud and the passage of the seeding plume between 10:36 - 10:41 UTC (indicated by the gray bar). Additionally shown are the flight level of the tethered balloon system (TBS, 10 s rolling mean, black line) and the seeding start time (pink star). **(b)**: Shown are the temporal evolution of cloud droplet number concentration (CDNC, 10 s rolling mean, solid blue) and ice crystal number concentration (ICNC, 10 s rolling mean, dashed orange) over the course of the seeding experiment. The gray bar indicates the expected passage of the seeding plume at the TBS location, which is slightly later than in **(a)**, since the TBS flies downwind of the main site (see Fig. 1). The median CDNC obtained from the background (bgd) is shown as dashed black lines, which also indicates the time periods used to determine the median. The liquid water content (LWC) vs. ICNC is shown in **(c)** and CDNC vs. ICNC is shown in **(d)**, with 1 s rolling mean applied to the data. The background median values for LWC and CDNC are shown in dashed black and linear regressions are applied (black lines) with Pearson correlation coefficients r$^2$ = 0.66 in **(c)**, and r$^2$ = 0.68 in **(d)**.

## 2.3 Observations during an exemplary seeding experiment

The observations of one seeding experiment (SE Nr. 3 at -5.5 °C and residence time of 8 min) performed on January 25, 2023 are shown in Fig. 2. The temporal evolution of the TBS height and radar reflectivity are shown in Fig. 2a and the evolutions of the CDNC and ICNC are shown in Fig. 2b. The cloud base is 1.1 km a.s.l. and the cloud top is $\approx$ 1.4 km a.s.l.. The unseeded background cloud has a mean reflectivity of $\approx$ -30 dBZ, a median CDNC of $\approx$ 410 cm$^{-3}$, and no ice crystals (Fig. 2a, b). During the passage of the seeding plume, the radar reflectivity increases by 20 dBZ, the ICNC reaches values of $\geq$ 600 L$^{-1}$

(10 s average), and the CDNC decreases by up to 75 %. A significant negative correlation is observed between the liquid water content (LWC) and the ICNC (Fig. 2c, Pearson $r^2 = 0.66$) as well as between the CDNC and the ICNC (Fig. 2d, Pearson $r^2 = 0.68$). These negative correlations between the liquid phase parameters and the ICNC are clear indicators of the WBF process, where the liquid phase is depleted by the ice phase. We would also like to emphasize that we observe single patches that are fully glaciated, i.e., where CDNC and LWC reach 0, after only $\approx$ 8 min (Fig. 2c, d).





## 2.4 Estimating residence times of ice crystals

The most important variable in determining accurate ice crystal growth rates is the ice crystal residence time, i.e., the time between the nucleation of ice crystals and their in situ observation. The most direct way to infer residence times is to use the seeding start time (i.e., the ignition of the seeding flare) and the time of first ice crystal appearance in the in situ data. However, this simplistic approach may lead to an overestimation, since the first part of the seeding plume may be missed by the in situ observations, since there is a possibility that the first part does not pass through HOLIMO. Therefore, we developed a method described in the following section to reduce the likelihood of missing the leading edge of the seeding plume. In Sect. 2.4.2, we discuss other potential sources of uncertainty that could lead to over- or underestimation of residence times and how to deal with them.

### 2.4.1 Retrieving residence times from in situ and remote sensing data

Our approach to address the potential overestimation in residence times, integrates all available additional remote sensing observations of the seeding plume. The remote sensing observations have a higher spatial coverage, which improves the chances of detecting the leading edge of the seeding plume. We first determine the observed wind speeds for all observations based on the locations and timestamps of the seeding and observations, respectively. We then use the maximum observed wind speed to inversely calculate the residence time for the in situ observation. The information on the locations and timestamps of seeding and observation is determined as follows:

- **Seeding start**: Seeding start time is defined as the time of flare ignition (see Table A1) and the seeding location as the center of the flown seeding legs. Both the start time and the respective seeding locations are provided directly by the UAV GPS data.

- **In situ**: For the in situ observations of HOLIMO, we use the time of first appearance of ice crystals as the observation time. The time is provided by the instrument's motherboard, which is synchronized to the cellular network, while the location is obtained by a GPS sensor.

- **Remote sensing**: The three different cloud radars were operated using combinations of vertical-pointing height-time indicator (HTI) measurements, partial-plan-position indicator (PPI) scans, i.e., at constant elevation and specified azimuth range (usually covering $90°$ centered around the wind direction), and partial-range height indicator (RHI) scans, i.e., variable elevation scanning through the zenith perpendicular to the wind direction (fixed azimuth). For the HTI and RHI measurements, the location is defined by the main site, and for the PPI we included a correction for horizontal displacement. The observation timestamps for all remote sensing measurements were manually visually determined and are based on the first significant increase in radar reflectivity ($\approx 5\text{-}10\,\mathrm{dBZ}$) and / or linear depolarization ratio ($\approx 10\text{-}15\,\mathrm{dB}$) compared to the radar reflectivity of the unseeded background cloud.



Based on these locations and times, we are able to calculate the wind speed for each observation. If the wind speeds derived from the different instruments differ significantly for a seeding experiment, this indicates that some instruments missed the first part of the seeding plume.

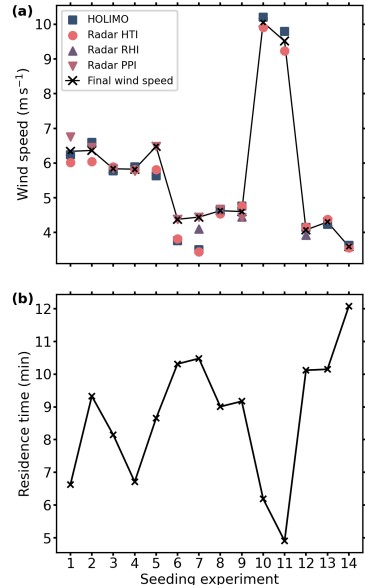

**Figure 3.** Wind speeds, residence times for all seeding experiments. **(a)**: Wind speeds for each seeding experiment determined from in situ measurements by HOLIMO (blue squares), remote sensing observations of height-time indicator (HTI) measurements (red circle), range-height indicator (RHI) scans (purple upward triangles), and plan position indicator (PPI) scans (pink downward triangles). The final wind speed estimates for each seeding experiment are shown as a solid black line with crosses. **(b)**: Calculated residence times for all seeding experiments using the wind speeds from **(a)** and the seeding distances from Table A1.

    The retrieved wind speeds for all measurements are shown in Fig. 3a, along with the final wind speed used to calculate the residence times for each seeding experiment. The final wind speed is determined as follows: If all derived wind speeds are

within 10 % of the HOLIMO wind speed, we assume to be within the respective uncertainties of our method and use an average value, e.g., as for seeding experiment 1. If any wind speed of the remote sensing observations is more than 10 % higher than the wind speed determined by HOLIMO, we assume that HOLIMO missed the first part of the seeding plume and we use the maximum wind speed observed for that seeding experiment, e.g., as for seeding experiment 5. These final wind speeds are then combined with the seeding distances between the seeding UAV location and the in situ measurement by HOLIMO (given in

Table A1) to inversely calculate the ice crystal residence times for each seeding experiment shown in Fig. 3b. These residence times will be used in Sects. 2.5 and 3 to calculate the ice crystal growth rates. The numerical values are given in Table A1.





### 2.4.2 Further sources of uncertainties in estimating residence times

Apart from overestimating the residence time by missing the leading part of the seeding plume (see Sect. 2.4.1), we want to address two other sources that can potentially lead to overestimation and underestimation of residence times.


1. We assume that ice crystals nucleate immediately after seeding. Any delay in nucleation or later nucleation events would lead to an overestimation of the residence time. The studies by Ramelli et al. (2024) and Miller et al. (2024a) suggest immediate nucleation of ice crystals on our time scales, but we still cannot rule out the occurrence of later nucleation events.

2. Our assumption, that the ice crystals are advected on a direct path (i.e., a straight line) between the seeding and measurement locations, may be violated. For example, some of the ice crystals may have taken a longer path due to turbulence. If these ice crystals mix back into the more directly advected seeding plume, we will underestimate their growth time. To reduce this overestimation, we only include ice crystals that are within $5.5\,\mathrm{min}$ (flare burning time) of their expected arrival based on the residence time calculated in Sect. 2.4.1, thus excluding those with exceptionally long pathways.

## 2.5 Calculating ice crystal growth rates

The vapor diffusional growth rates of ice crystals are calculated by dividing the major axis length of pristine ice crystals by the residence time estimated in Sect. 2.4.1. Our seeding temperatures range between -5.1 °C and -8.3 °C and thus mainly fall in the columnar growth regime (Nakaya, 1954; Takahashi et al., 1991). Only the coldest one at -8.3 °C (seeding experiment 12), we observed a mixture of plates and columns, since the growth rates along the basal plane and prism plane are approximately
equal in this temperature regime (Takahashi et al., 1991). We will only report ice crystal growth rates along the major axis as these are less affected by measurement inaccuracies. We use the following assumptions for our calculations:

- The environmental conditions and microphysical properties of the background cloud remain constant throughout a seeding experiment.

- The ice crystals grow linearly over time, i.e., the growth rate is constant.

- The ice crystals nucleate immediately after the seeding agent is released.

- The ice crystals were advected on a straight path from the seeding to the measurement location.

- The burning time of the flare is $5.5\,\mathrm{min}$.

The latter three assumptions and their implication were discussed in detail in Sect. 2.4.2. In the following analysis, we will only consider the growth rates of pristine ice crystals.





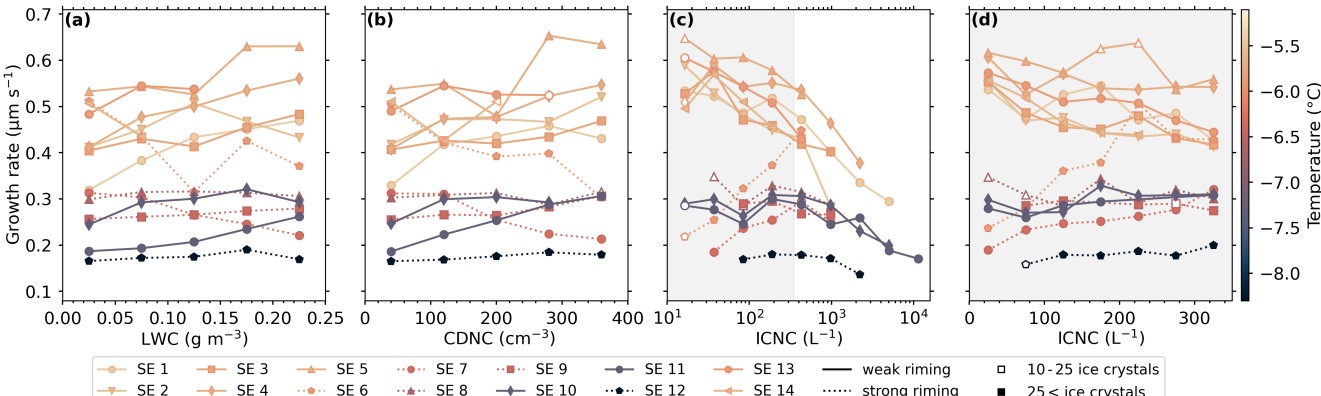

**Figure 4.** Vapor diffusional ice crystal growth rates for the 14 seeding experiments (i.e., SE 1) versus liquid water content (LWC) in **(a)**, cloud droplet number concentration (CDNC) in **(b)**, and ice crystal number concentration (ICNC) with logarithmic scaling in **(c)** and linear scaling in**(d)** for a selected range (gray shading in **(c)**). Vapor diffusional growth rates are shown as mean values of the respective bins and are plotted at the bin centers with bin sizes of $0.05\,\mathrm{g\,m^{-3}}$ for the LWC, $80\,\mathrm{cm^{-3}}$ for the CDNC, 10 logarithmically scaled bins for the logarithmic-plotted ICNC, and $50\,\mathrm{L^{-1}}$ for the linear-plotted ICNC. Values are shown only if > 10 ice crystals fall in the bin, and unfilled markers indicate bins with 10 - 25 ice crystals. Experiments with weak or no riming are shown as solid lines and experiments with strong riming are shown as dotted lines. The color indicates the temperature of each seeding experiment.

## 3 Vapor diffusional ice crystal growth rates in natural clouds

The vapor diffusional growth rates for all seeding experiments are shown in Fig. 4 as mean values of the bins versus the LWC, CDNC, ICNC. First, we see that the seeding temperature (color scale) strongly influences the diffusional growth rates. The maximum growth rates from our seeding experiments were found at about -5.3 °C with values $> 0.6\,\mathrm{\mu m\,s^{-1}}$, whereas the lowest growth rates were observed at about -8.3 °C with values $< 0.2\,\mathrm{\mu m\,s^{-1}}$. This decrease in growth rates with temperature is consistent with the general theory of ice crystal growth habits (see Nakaya, 1954). Maximum columnar growth rates along the c-axis are expected around -5 °C and reach a minimum at -15 °C. At around -8 °C, we enter the transition regime to plate-like growth, which dominates for temperatures between -10 and -22 °C (Nakaya, 1954; Takahashi et al., 1991).

To assess the relationship between ice crystal growth by vapor diffusion and cloud microphysical variables, we need to distinguish between seeding experiments with weak or no riming (solid lines in Fig. 4, hereafter referred to as weak riming) and seeding experiments with strong riming (dotted lines in Fig. 4), which was visually confirmed during the manual classification of ice crystals. The vapor diffusional ice crystal growth rates observed in the weak riming experiments generally decrease with decreasing LWC and CDNC (Figs. 4a, 4b) as expected (Song and Lamb, 1994; Takahashi, 2014), and with increasing ICNC (Fig. 4c). The trends of the weak riming experiments indicate that the diffusional ice crystal growth rates in our seeding experiments are influenced not only by temperature, but also by the microphysical properties of the cloud, which is not surprising. For lower LWC and CDNC, the depletion of the vapor phase due to diffusional ice crystal growth will be slower replenished



by the evaporation of cloud droplets, thereby slowing down the diffusional growth of ice. Further, in our seeding experiments, we produce exceptionally high ICNCs for our cloud type (low stratus, non-convective), well above $> 100\,\mathrm{L}^{-1}$ (generally no ice crystals were observed in the unseeded low stratus cloud). Such conditions may naturally be found during secondary ice production events, which can causes strong inhomogeneities of the liquid and ice phase within MPCs. For these high ICNCs

conditions, we expect a rapidly glaciating MPC, which will lead to a "competition" for the available water vapor among the ice crystals as described by Ramelli et al. (2024). For high ICNC, we additionally expect a size-dependent sampling bias towards smaller ice crystals due to aggregation and inherently lower growth rates, which will be discussed in Sect. 3.1. Thus, for our seeding experiments, the ICNC is the second most important factor in controlling the vapor diffusional growth rates of ice crystals, following temperature, as shown by the strong decrease in growth rates with increasing ICNC (Fig. 4c). The strong

riming experiments show either a less pronounced or even counterintuitive trend with LWC, CDNC, and ICNC, which we think mainly arises from a size-dependent sampling bias that will be discussed in more detail in Sect. 3.1.

To also report vapor diffusional ice crystal growth rates that more closely resemble naturally occurring and not rapidly glaciating MPCs, we present strategies to reduce the direct implications of high ICNC on ice crystal growth and how to minimize the size-dependent sampling biases. We first address the sampling biases (Sect. 3.1) before, we present two individually applied

selective filtering approaches: the first approach uses thresholds based on microphysical variables to restrict our data set to more favorable (non rapidly glaciating) growth conditions (Sect. 7), and the second approach is more stochastic in nature and depends only on the ice crystal size (Sect. 3.3). These filtering approaches will also make our growth rate data set more comparable with laboratory studies, which is done in Sect. 3.4, where we report the the unfiltered and filtered ice crystal growth rates versus temperature.

## 3.1 Aggregation-, riming-, and sedimentation-induced sampling biases on ice crystal growth rates

In this study, we report vapor diffusional growth rates of *pristine* ice crystals, which means that we exclude ice crystals from our analysis that have grown other processes such as riming and aggregation. This separation, however, introduces a size-dependent sampling bias, addressed in the following.

The riming efficiency of ice crystals strongly depends on the ice crystal size (Wang and Ji, 2000) and CDNC and is highest for

large ice crystals and high CDNC. This means that a relatively higher number of large ice crystals will grow by riming, and by excluding them from our analysis, we introduce a bias towards smaller ice crystals and therefore lower growth rates, which is visualized in Fig. 5. The strong dependence of riming on ice crystal size and its connection to the liquid phase makes it challenging to remove this bias using a filtering approach with thresholds based on cloud microphysical variables, as described in Sect. 3.2. Therefore, we decided to exclude the seeding experiments with strong riming from our analysis using the threshold

based approach.

Similar to riming, aggregation also introduces such a bias towards smaller diffusional growth rates due to its size dependence, but this bias can be more easily accounted for. Since the temperature is constant during seeding experiments, the aggregation efficiency is primarily controlled by the ICNC and to a lesser extent by the ice crystal size (Hobbs et al., 1974): the aggregation efficiency should be at maximum for high ICNC and large ice crystals. Therefore, we aim to reduce this bias by limiting our





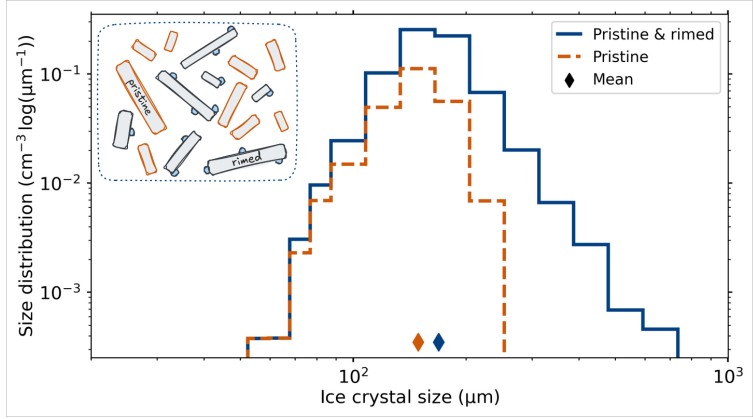

**Figure 5.** Ice crystal size distributions from seeding experiment 9 of rimed and pristine crystals (blue) and of pristine crystals only (orange), with diamonds indicating the mean ice crystal size. The inset drawing illustrates the higher likelihood of large ice crystals to be rimed.

data set to observations with low ICNC (see Sect. 3.2), which we do anyway to make our results more comparable to non-rapidly glaciating MPCs.

Finally, there is also a sampling bias toward smaller ice crystals due to sedimentation. Larger rimed ice crystals tend to have a lower surface-to-volume ratio and thus a higher fall velocity. However, due to the dispersion of our seeding plume, which typically form ice crystals over the full vertical extent of the seeded cloud (see Fig. 2, and Henneberger et al. (2023); Ramelli

et al. (2024); Miller et al. (2024a)), we expect this bias to have a small impact only, as a similar number of large ice crystals will settle into our sampling volume from above as will sediment out below.

### 3.2 Approximating unconstrained diffusional ice crystal growth conditions in MPCs

The high ICNC observed in our seeding experiments may lead to rapidly glaciating MPCs, with highly or even fully glaciated patches, where ice crystal growth by vapor diffusion is limited due to lack of available water vapor. In this filtering approach,

the goal is to find suitable thresholds in LWC, CDNC, and ICNC to restrict our data set to conditions with unconstrained (high) ice crystal growth rates. However, the margin of these restrictions is limited, as it reduces the number of available data points, i.e., the number of pristine ice crystals, particularly by restricting the ICNC. The seeding experiments with strong riming (see Sect. 3.1 and table A1) are excluded from this analysis.

To determine general thresholds of LWC, CDNC, and ICNC, for all seeding experiments ($LWC_{th}$, $CDNC_{th}$, and $ICNC_{th}$),

we first need to remove the temperature dependence by normalizing all growth rates by the respective mean growth rate of each seeding experiment. The resulting normalized growth rate distributions are shown in Fig. 6 (blue shadings) vs. LWC, CDNC, and ICNC, with mean trends of their dependencies indicated by the white circles. For LWC (Fig. 6a), we find a positive linear correlation with the growth rates, and for CDNC (Fig. 6b), we observe an initial increase in growth rates with increasing CDNC before reaching a plateau region for $CDNC \geq 100 \, cm^{-3}$. The diffusional growth rates vs. ICNC (Fig. 6c) are characterized by a





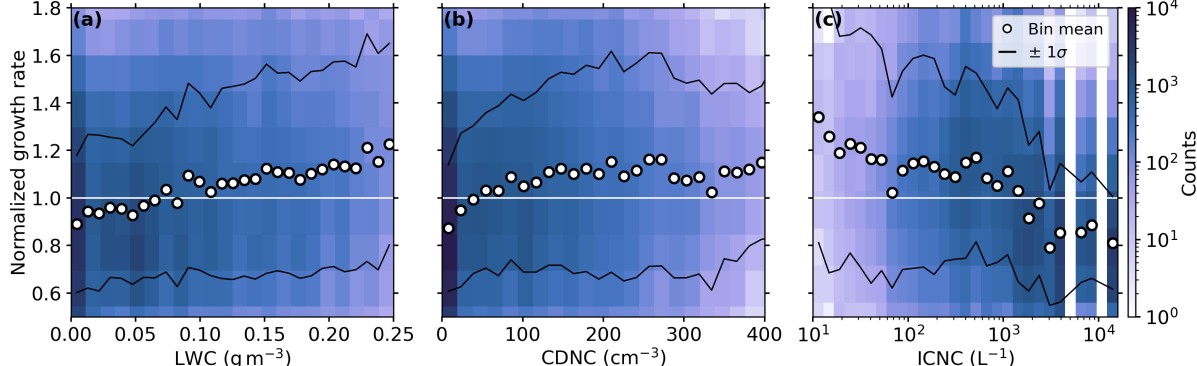

**Figure 6.** Ice crystal growth rate distributions (color-coded in counts per bin) from all weak riming seeding experiments (see table A1), where the rates are normalized by the respective mean growth rate of each seeding experiment, vs. liquid water content (LWC) in **(a)**, cloud droplet number concentration (CDNC) in in **(b)**, and ice crystal number concentration (ICNC) in **(c)**. The mean growth rates of each vertical bin column (white circles) and the corresponding $\pm 1\sigma$ standard deviations (solid black lines) are shown to better highlight trends in the distributions. The 1-line (solid white) is shown for better comparability.

continuous decrease with increasing ICNC. These trends reflect the findings that were already observed in Fig. 4, however in a more generalized form for all seeding experiments together. To achieve unconstrained vapor diffusional growth conditions, our microphysical thresholds should limit our data set to conditions with high LWC, high CDNC and low ICNC. After applying adequate thresholds, we expect to observe no remaining dependencies, meaning the diffusional ice crystal growth rates will remain constant with respect to the microphysical variables.

From the trends in Fig. 6b, we see that for $\text{CDNC} \leq 100\,\text{cm}^{-3}$ the diffusional growth rates are significantly reduced, while for $\text{CDNC} \geq 100\,\text{cm}^{-3}$ no trend is visible. This feature indicates a suitable $\text{CDNC}_{\text{th}}$ to constrain our data set. For the LWC, no such clear threshold can be identified because the growth rate trends persist over the entire LWC range. Since the LWC and CDNC are closely connected we thus we will discard an LWC-based threshold. For the ICNC, no clear $\text{ICNC}_{\text{th}}$ can be inferred either. Although we cannot identify a distinct threshold, it is evident that the vapor diffusional ice crystal growth is limited at

high ICNC as shown in Ramelli et al. (2024) and discussed in the previous sections. Based on that, we apply an $\text{ICNC}_{\text{th}}$ to limit our data set to lower ICNCs. Applying an $\text{ICNC}_{\text{th}}$ is also crucial to reduce the aggregation bias discussed in Sect. 3.1. As lower bound, we define an $\text{ICNC}_{\text{th}} \geq 80\,\text{L}^{-1}$ to ensure a sufficient number of pristine ice crystals for statistical reasons.

Generally, it is important to note that applying a variable-specific threshold to the same variable will not alter its general trend, but only trim the respective range. Thus, the growth rate trends are then defined by the remaining data set and will follow the

respective trends observed in Fig. 6.

After having defined upper and lower bounds, we systematically adjust the values for $\text{CDNC}_{\text{th}}$ and $\text{ICNC}_{\text{th}}$ in a gradual manner, i.e., incrementally increasing $\text{CDNC}_{\text{th}}$ from 25 to $125\,\text{cm}^{-3}$ and decreasing $\text{ICNC}_{\text{th}}$ from 400 to $80\,\text{L}^{-1}$. To better understand their individual effects on the data, we first apply each threshold independently before testing combinations of the





two. To evaluate the respective effectiveness of applying the thresholds, and to better visualize the implied changes, we will

use linear fits for LWC and CDNC, and exponential fits for ICNC, since linear fits were found to be ineffective for the latter (see Table B1). To also incorporate a weighting based on frequency of occurrence, all fits are applied to the normalized, and non-averaged ice crystal growth rates (histogram data, blue shadings, in Fig. 6).

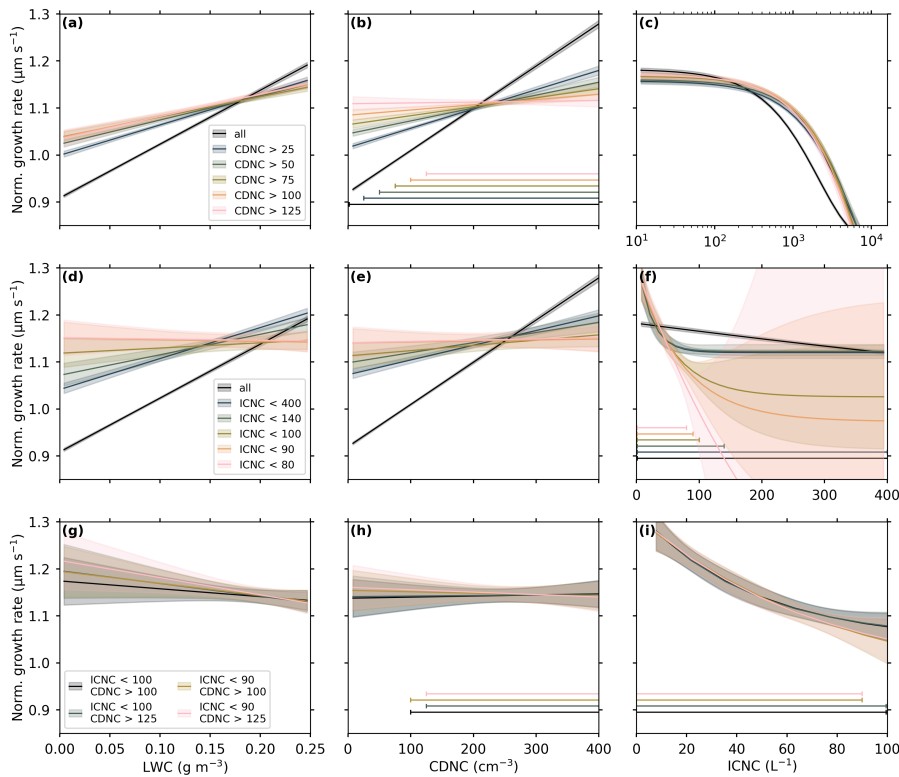

**Figure 7.** Changes in the dependencies of diffusional ice crystal growth rates on cloud microphysical variables after applying a gradually increasing cloud droplet number concentration threshold ($\text{CDNC}_{\text{th}}$) in **(a)**-**(c)**, a gradually decreasing ice crystal number concentration threshold ($\text{ICNC}_{\text{th}}$) in **(d)**-**(f)**, and a combination of $\text{CDNC}_{\text{th}}$ and $\text{ICNC}_{\text{th}}$ in **(g)**-**(i)**. The changes are visualized by linear fits for liquid water content (LWC) in **(a)**,**(d)**, and **(g)** and cloud droplet number concentration (CDNC) in **(b)**, **(e)**, and **(h)**, and by exponential fits for ice crystal number concentration (ICNC) in **(c)**, **(f)**, and **(i)** together with their 95 % confidence intervals (shaded regions). The horizontal bars indicate the used range after applying the respective thresholds. The growth rate distributions (similar to Fig 6) of all individual steps are shown in Figs. B1 - B3 and fit parameters are given in Tables B1.

The resulting changes in dependencies between growth rates and microphysical variables after applying $\text{CDNC}_{\text{th}}$ and $\text{ICNC}_{\text{th}}$ are shown in Fig. 7. In Fig. 7a - c, a total of five $\text{CDNC}_{\text{th}}$, varying between 25 and 125 $\text{cm}^{-3}$, are applied to our

data set. The trimmed diffusional growth rate distributions after applying the $\text{CDNC}_{\text{th}}$ are shown in Fig. B1 and the corresponding fit parameters are given in Table B1. The growth rates become increasingly independent of LWC, with increasing $\text{CDNC}_{\text{th}}$, as seen by the linear fits (slopes are less steep), until a $\text{CDNC}_{\text{th}} = 75\,\text{cm}^{-3}$. For higher $\text{CDNC}_{\text{th}}$, no significant



changes are observed for the LWC (Fig. 7a). For the CDNC in Fig. 7b, we observe the expected behavior, reaching a diffusional growth regime which is mainly unaffected by the CDNC for $\text{CDNC}_{\text{th}} \geq 100\,\text{cm}^{-3}$, with slopes approaching zero. The

growth rate dependence on ICNC is largely unaffected by the application of $\text{CDNC}_{\text{th}}$, and no significant changes are observed for $\text{CDNC}_{\text{th}} \geq 25\,\text{cm}^{-3}$ (Fig. 7c).

Five different $\text{ICNC}_{\text{th}}$ between 400 and $80\,\text{L}^{-1}$ are used to restrict our data set to low ICNCs in Fig. 7d - f. The trimmed diffusional growth rate distributions vs. LWC, CDNC, and ICNC and the fitting parameters are shown in Fig. B2 and Table B1. In Fig. 7d and e, we see that the respective diffusional growth rates are gradually less affected by the LWC and CDNC with

decreasing $\text{ICNC}_{\text{th}}$, visible by the slopes of the linear regressions approaching zero. For all $\text{ICNC}_{\text{th}} \leq 100\,\text{L}^{-1}$, no significant change in the slopes can be inferred, as they are all within their respective uncertainties (see Table B1). The diffusional growth rate dependence on ICNC approaches the initial strong decrease observed in Fig. 6c for $\text{ICNC} < 100\,\text{L}^{-1}$.

Fig. 7g - i show how combinations of $\text{CDNC}_{\text{th}}$ and $\text{ICNC}_{\text{th}}$ (permutations of $\text{CDNC}_{\text{th}} \in \{\geq 100; \geq 125\}\,\text{cm}^{-3}$ and $\text{ICNC}_{\text{th}} \in \{\leq 100; \leq 90\}\,\text{L}^{-1}$) affect the trends of the growth rate. The diffusional growth rate distributions for the combinations are

shown in Fig. B3 and the fit parameters are given in Table B1. No significant differences in the slopes can be inferred from the given fit parameters, as all fit parameters are within their respective uncertainties. Further, the flip in slopes for the LWC to negative values (Fig. 7g) indicates that we have reached the noise level of our data set by reducing the number of data points (see Fig. B3). Since no significant changes can be inferred using more conservative combinations, we decide to proceed with a $\text{CDNC}_{\text{th}} = 100\,\text{cm}^{-3}$ and an $\text{ICNC}_{\text{th}} = 100\,\text{L}^{-1}$ to report vapor diffusional growth rates for unconstrained ice crystal growth

conditions. These more liberal thresholds also help to solidify the statistical aspect of our study by maintaining higher absolute numbers of pristine ice crystals for each seeding experiment (see Table A1). These microphysical thresholds will be applied on our dataset in Sect. 3.4.

### 3.3  The "lucky ice crystal" approach

The use of the threshold-based approach described in the previous section helps to approximate unconstrained ice crystal growth conditions by reducing the dependence of growth rates on LWC and CDNC, but the dependence on the ICNC still remains. Therefore, we want to introduce another filtering approach that is more stochastic in nature and will be applied independently of the microphysical threshold-based approach on our growth rate data set.

The idea behind this approach is to select the "lucky ice crystals" that experience the most favorable growth conditions, i.e.,

highest average supersaturation, allowing them to grow to the largest sizes. Since this approach is independent of LWC, CDNC, and ICNC, we do not expect it to be affected by the sampling biases introduced in Sect. 3.1, which allows us to include the seeding experiments with strong riming (see Table A1). We use the $80^{\text{th}}$ percentile as a lower limit for the growth rates, determined individually for each seeding experiment. This threshold was chosen for statistical reasons to maintain the number of pristine ice crystals per experiment $\geq 100$ (see Table A1). This approach will be applied in Sect. 3.4 on our data set.



### 3.4 Temperature dependency of vapor diffusional ice crystal growth rates and comparison to laboratory studies

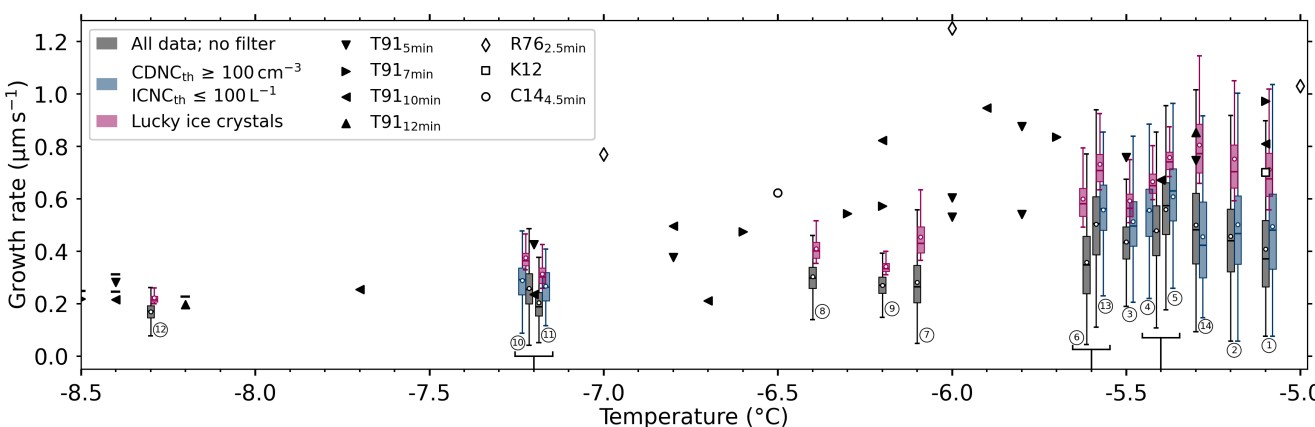

**Figure 8.** Diffusional growth rates along the major axis of pristine ice crystals versus temperature as box plots (lower quartile and upper quartile; whisker: 1.5 interquartile range; median: horizontal line, mean: white circle). Growth rates are determined from the seeding experiments (encircled numbers, see Table A1) using no filter / the raw data set (gray), using filtering with thresholds based on the cloud droplet number concentration $CDNC_{th} \geq 100\,L^{-1}$ and on the ice crystal number concentration $ICNC_{th} \leq 100\,L^{-1}$ (see Sect. 3.2) (blue), and selecting only the largest ice crystal ("lucky ice crystals", $80^{th}$ percentile, red, Sect. 3.3). The five strong riming experiments are excluded from the threshold based approach. Seeding experiments at identical temperatures, i.e. (4, 5), (6, 13), and (10, 11), are shifted slightly for better visibility, with their actual temperatures indicated by the black brackets below. The markers denote growth rates reported in laboratory studies of Takahashi et al. (1991, raw data provided) as T91, Ryan et al. (1976, Figure 4) as R76, Knight (2012, Figure 7a) as K12, and Castellano et al. (2014, Figure 5a) as C14. Where times are shown as suffixes, the growth rates were derived from the ice crystal size and the corresponding growth time of the laboratory study. The T91 growth rates below -8°C (indicated by a bar) are mean values of the basal and prism face growth rates.

The diffusional growth rates along the major axis of pristine ice crystals versus temperature for the 14 seeding experiments of this study are shown in Fig. 8, together with growth rates from laboratory-based studies of Ryan et al. (1976), Takahashi et al. (1991), Knight (2012), and Castellano et al. (2014). For each seeding experiment, we report a distribution of diffusional growth rates based on the three previously described filtering approaches: 1) unfiltered ice crystal data; 2) filtering based on thresholds of cloud microphysical variables ($CDNC_{th} \geq 100\,cm^{-3}$ and $ICNC_{th} \leq 100\,L^{-1}$, see Sect. 3.2); and 3) lucky ice crystal approach, with filtering based on ice crystal sizes ($80^{th}$ percentile, see Sect. 3.3).

The trends of the growth rates with temperature show the expected behavior for columnar growth along the basal face (Nakaya, 1954), reaching highest values around -5.3 °C and decreasing towards colder temperatures. This indicates the transition to the plate-like growth regime around -8 °C, where basal-and prism-face growth rates are approximately equal. The unfiltered diffusional growth rates (gray) are the lowest and generally show the largest deviations from the laboratory data, especially for seeding experiments 7, 8, and 9 in the range between -6.0 and -6.5 °C. The growth rates determined by applying thresholds of cloud microphysical variables (red) show ≈ 13 % higher values (except for seeding experiment 14) compared to the unfiltered





one. Unsurprisingly, the highest vapor diffusional growth rates are found using the "lucky ice crystal" approach ($\approx 48\,\%$ higher compared to unfiltered), which also shows the closest resemblance to laboratory-based studies.

In general, our vapor diffusional growth rates derived from in situ measurements in natural clouds tend to be lower than the growth rates observed in the laboratory. The largest differences occur between the unfiltered data set and laboratory studies. The ice crystal growth rates derived using the threshold-based filtering approach, which aims to approximate more unconstrained growth conditions, better resemble the laboratory data but still remain significantly lower. These discrepancies likely arise from the very different growth conditions experienced by ice crystals in laboratory versus field measurements. In contrast to our field

measurements, laboratory studies usually have an unlimited water vapor reservoir and often significantly higher LWCs, e.g., $\mathrm{LWC} \geq 1\,\mathrm{g\,m}^{-3}$ in Ryan et al. (1976) and in Castellano et al. (2014), which promotes increased vapor diffusional ice crystal growth rates. Furthermore, the growth rates of laboratory studies are usually based on individual and consecutively grown ice crystals, which experience no interference or competition for water vapor from other ice crystals. To better reconcile in situ and laboratory data, future laboratory experiments could use setups with limited water vapor supply and investigate the growth

rates in environments with multiple ice crystals. Future seeding experiments, on the other hand, should consider seeding with lower seeding particle concentrations, e.g., using smaller flares or distributing the seeding particles over larger areas. The ice crystal growth rates derived using the threshold based filtering approach to approximate more unconstrained growth conditions better resemble the laboratory data, but are still significantly lower. The best agreement between in situ and laboratory data is found using the lucky ice crystal approach, with laboratory values still slightly higher.

Given the significant differences between in situ and laboratory conditions, we do not want to make a definitive statement about "over- or underestimation" of natural diffusional ice crystal growth rates. Instead, we would like to emphasize that we can reach a reasonable agreement, despite the large differences in measurement conditions between the seeding experiments and laboratory studies.

    Based on these findings, we suggest treating the ice crystal growth rates determined from the unfiltered data set as a lower

bound for natural clouds or valid for seeding experiment conditions. The growth rates derived using the microphysical threshold based approach likely best resembles situations with high secondary ice production in natural clouds and rapidly glaciating MPCs. The growth rates retrieved from the lucky ice crystal approach should be treated as an upper bound for natural cloud and are most representative for comparisons with laboratory studies.

## 4   Conclusions

In this study, we conducted glaciogenic cloud seeding experiments in persistent supercooled low stratus clouds. The novelty of our experimental design, which emphasizes high controllability and repeatability, was key to retrieve the vapor diffusional growth rates of pristine ice crystals from in situ observations of 14 seeding experiments in the temperature range of -5.1 °C to -8.3 °C. The observed temperature dependence of linear growth rates peaks around -5.5°C and decrease toward colder temperatures, which aligns with previous laboratory studies.




The observed vapor diffusional ice crystal growth rates were primarily controlled by the temperature, with LWC, CDNC and ICNC being a secondary controlling factor. The diffusional growth rates increase with LWC, initially increase with CDNC before leveling of at CDNC $\geq 100\,\mathrm{cm}^{-3}$, and decrease with ICNC.

The seeding experiments in this study generally reach significantly higher ICNC ($\geq 100\,\mathrm{L}^{-1}$) than found in natural stratiform clouds, which can lead to rapidly glaciating MPCs. Under these conditions, ice crystal growth is reduced due to competi-

tion among the ice crystals for the available water vapor, as shown in Ramelli et al. (2024). Therefore, we introduced two independent approaches to filter our data set to better approximate unconstrained ice growth conditions, less affected by our experimental design. The first approach used thresholds based on cloud microphysical variables (CDNC $\geq 100\,\mathrm{cm}^{-3}$ and ICNC $\leq 100\,\mathrm{L}^{-1}$), yielding on average $13\,\%$ higher growth rates than the unfiltered data set. The second approach selects the lucky ice crystals, i.e., the $20\,\%$ with the fastest growth rates, which yields on average $48\,\%$ higher growth rates than the

unfiltered data set.

    The vapor diffusional ice crystal growth rates inferred via the three different approaches (unfiltered, threshold-based, and lucky ice crystal) are reported as a function of temperature and are compared with laboratory measurements from Ryan et al. (1976); Takahashi et al. (1991); Knight (2012), and Castellano et al. (2014). The temperature dependence of our growth rates shows reasonable agreement with laboratory studies, although the absolute values tend to be lower for our in situ observations.

This discrepancy likely arises from differences in growth conditions: laboratory setups typically provide unlimited water vapor, high LWC, and isolated ice crystal growth, whereas our in situ observations involve limited water vapor availability and high ICNC.

We suggest treating the growth rates from the unfiltered data set, having largest deviation from the laboratory data, as lower bounds for natural growth rates or as those valid for conditions seen during seeding experiments. The threshold based filter

approach, most likely represents rapidly glaciating MPCs, e.g., regions of high secondary ice production with ICNC $\leq 100\,\mathrm{L}^{-1}$. The lucky ice crystal approach can be seen as an upper bound for ice crystal growth rates in natural clouds, with best aggreement to laboratory studies.

To better reconcile laboratory and field measurements, future seeding experiments should aim for lower ICNC, e.g. by releasing fewer seeding particles, using smaller flares, or laboratory studies should investigate growth under water-limited conditions and

environments with multiple ice crystals. Future cloud seeding experiments should also extend to warmer and colder temperature regimes beyond those investigated in this studied here.

    In summary, this study demonstrates how glaciogenic cloud seeding experiments can quantify vapor diffusional ice crystal growth rates in natural clouds, extending the work of Ramelli et al. (2024) across a broader temperature range and varying cloud microphysical conditions. Our results highlight the intricate relationships between diffusional ice growth and cloud

microphysical variables. These observation offer valuable insights into the complexity and variability of ice crystal growth in natural clouds, helping to refine laboratory studies, improve seeding experiments, and to validate theoretical ice growth models.

*Code and data availability.* Data and scripts will be uploaded into a repository upon acceptance, and are available upon request until then.





# Appendix A: Experiments overview



**Table A1.** Detailed overview of the 14 seeding experiments included in this study. Shown are the seeding experiment number used in this study; the CLOUDLAB mission ID; the seeding start time, i.e. ignition of the flare, in UTC; the temperature at seeding height in °C measured by the UAV; seeding distance in m between the UAV seeding location and the TBS; the estimated wind speed in $\mathrm{m\,s^{-1}}$ (see Sect. 2.4.1); the ice crystal residence time in s from the UAV seeding location to the TBS; the total number of observed ice crystals; the total number of pristine ice crystals; the number of pristine ice crystals after applying microphysical thresholds (Sect. 3.2); the number of "lucky" pristine ice crystals (Sect. 3.3).

| SE Nr. | Mission ID | Seeding start (UTC) | Temp. (°C) | Seeding distance (m) | Wind speed ($\mathrm{m\,s^{-1}}$) | Residence time (s) | Nr. ice crystals | Nr. pristine ice crystals all | thresholds | lucky ones |
|---|---|---|---|---|---|---|---|---|---|---|
| 1 | SM055 | 24/01/23 18:50:13 | -5.1 | 2521 | 6.3 | 398 | 23465 | 17867 | 1242 | 3575 |
| 2 | SM056 | 24/01/23 19:18:33 | -5.2 | 3562 | 6.4 | 560 | 10817 | 6663 | 2000 | 1388 |
| 3 | SM058 | 25/01/23 10:28:06 | -5.5 | 2853 | 5.8 | 489 | 4585 | 2344 | 233 | 469 |
| 4 | SM059 | 25/01/23 10:50:43 | -5.4 | 2346 | 5.8 | 403 | 6853 | 3448 | 192 | 759 |
| 5 | SM060 | 25/01/23 11:15:25 | -5.4 | 3363 | 6.5 | 519 | 1058 | 499 | 131 | 100 |
| 6* | SM061 | 25/01/23 18:55:35 | -5.6 | 2710 | 4.4 | 619 | 2116 | 539 | N/A | 126 |
| 7* | SM062 | 25/01/23 19:48:06 | -6.1 | 2793 | 4.4 | 629 | 9230 | 2189 | N/A | 545 |
| 8* | SM063 | 26/01/23 10:22:18 | -6.4 | 2502 | 4.6 | 540 | 3273 | 739 | N/A | 156 |
| 9* | SM064 | 26/01/23 10:48:33 | -6.2 | 2533 | 4.6 | 550 | 3910 | 1202 | N/A | 215 |
| 10 | SM074 | 27/01/23 16:00:00 | -7.2 | 3734 | 10.1 | 371 | 34503 | 31352 | 301 | 6413 |
| 11 | SM075 | 27/01/23 16:25:04 | -7.2 | 2801 | 9.5 | 294 | 32211 | 31404 | 186 | 6224 |
| 12* | SM096 | 09/01/24 09:44:20 | -8.3 | 2473 | 4.1 | 607 | 6678 | 1856 | N/A | 371 |
| 13 | SM104 | 12/01/24 09:03:22 | -5.6 | 2621 | 4.3 | 609 | 4060 | 1026 | 159 | 229 |
| 14 | SM105 | 12/01/24 09:37:44 | -5.3 | 2605 | 3.6 | 724 | 2165 | 835 | 61 | 117 |

*: Strong riming experiments; excluded from the threshold based filtering approach.



## Appendix B: Filtering using thresholds based on cloud microphysical variables

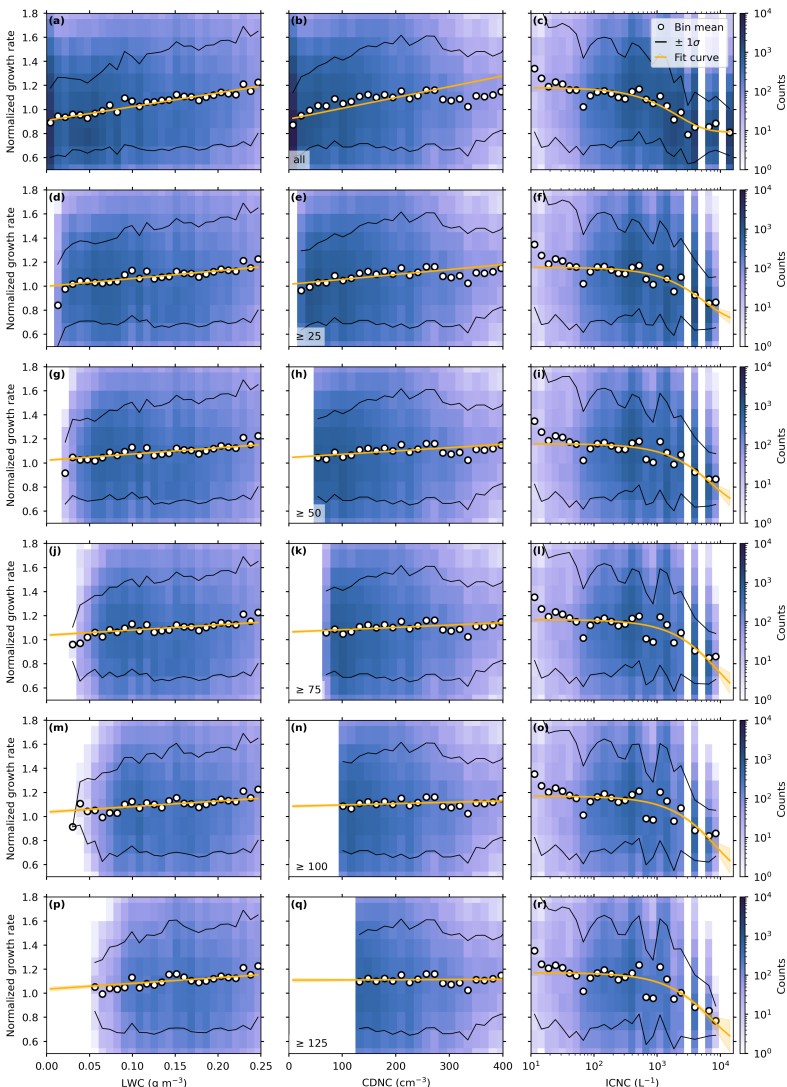

**Figure B1.** Ice crystal growth rate distributions (color-coded in counts per bin) from all weak riming seeding experiments (see table A1), where the rates are normalized to the respective mean growth rate of each seeding experiment. Five gradually increasing (top to bottom) cloud droplet number concentration thresholds (CDNC$_{th}$, given in the lower left corner of middle column) are applied on the data set and shown vs. the liquid water content (LWC) in in left column, the cloud droplet number concentration (CDNC) in the middle column, and ice crystal number concentrations (ICNC) in the right column. The mean growth rates (white circles) and respective $\pm 1\sigma$ standard deviations (solid black lines) for each column of bins are shown to better highlight trends of the distributions. Linear fits are used for the LWC and CDNC and exponential fits for the ICNC (solid orange; 95% confidence interval shading). The fits are based on the normalized ice crystal growth rate distributions to include a frequency-of-occurrence based weighting and fit parameters are given in Table B1.



Ice crystal growth rate distributions (color-coded in counts per bin) from all weak riming seeding experiments (see table A1)
, where the rates are normalized to the respective mean growth rate of each seeding experiment, vs. liquid water content (LWC)
in **(a)**, cloud droplet number concentration (CDNC) in in **(b)**, and ice crystal number concentration (ICNC) in **(c)**. The mean
growth rates of each vertical bin column (white circles) and the corresponding $\pm 1\sigma$ standard deviations (solid black lines) are
shown to better highlight trends in the distributions. The 1-line (solid white) is shown for better comparability.



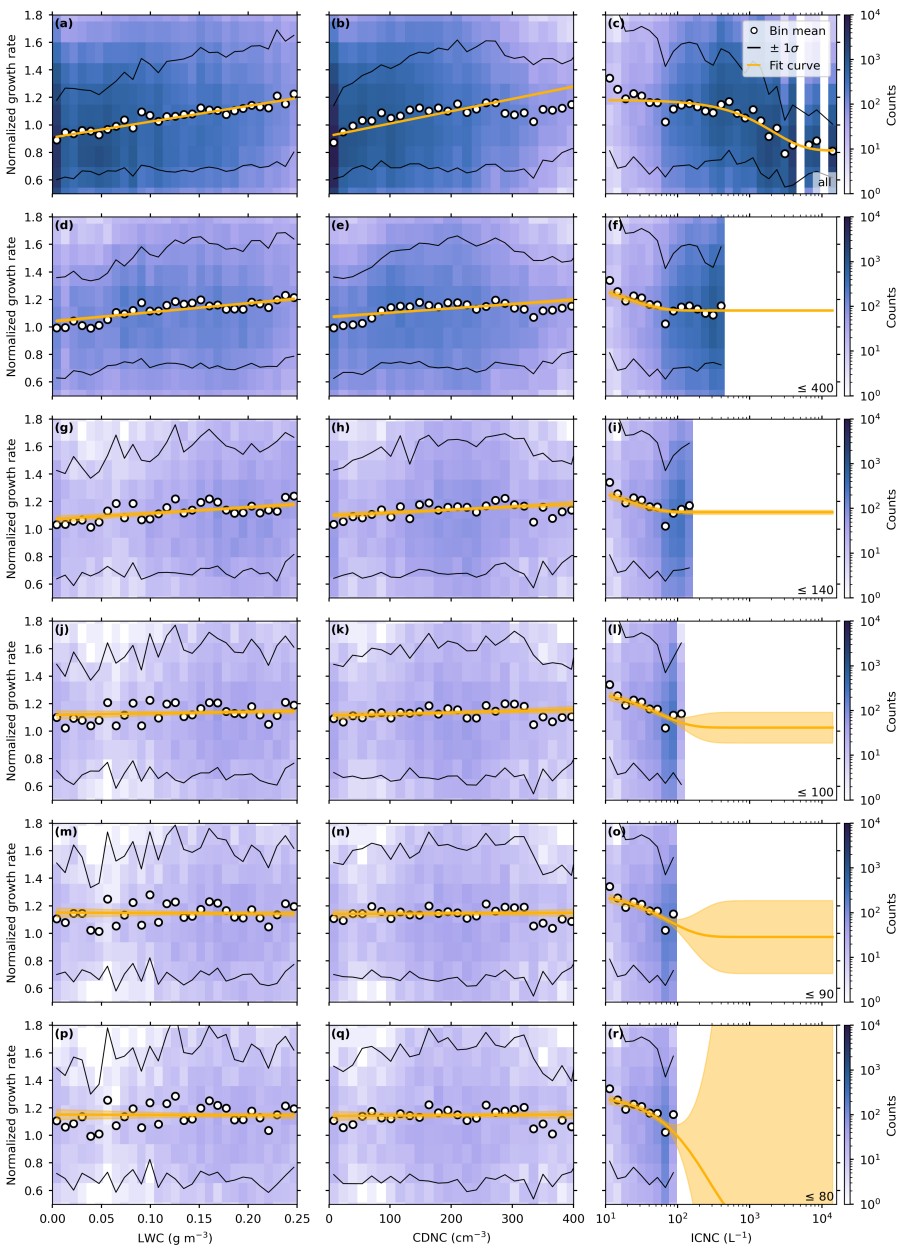

**Figure B2.** Identical to Fig. B1 but with five gradually decreasing (top to bottom) ice crystal number concentration thresholds (ICNC$_{th}$), given in the lower right corner of right column.



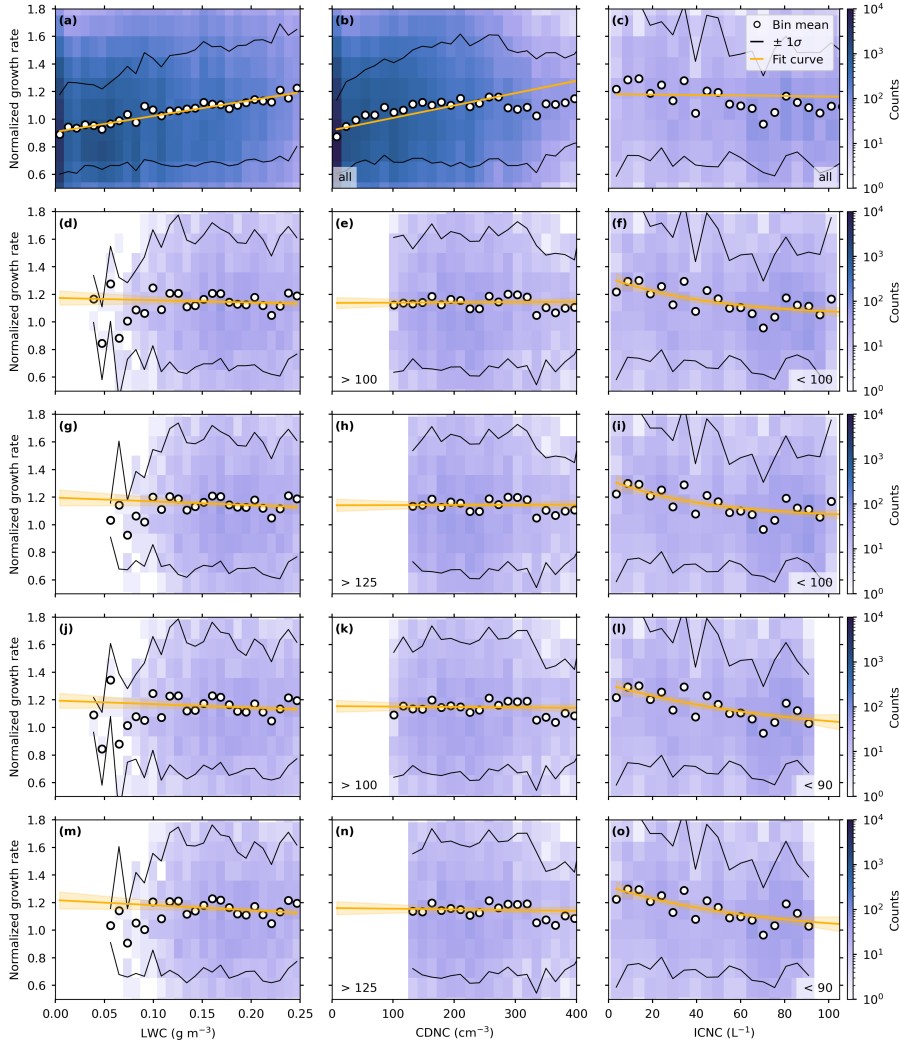

**Figure B3.** Identical to Figs. B1 and B2 but using a combination of cloud droplet number concentration thresholds ($CDNC_{th}$), given in the lower left corner of the middle column and ice crystal number concentration thresholds ($ICNC_{th}$), given in the lower right corner of right column.



**Table B1.** Fit parameters retrieved of the linear curves ($f(x) = m \cdot x + b$) for the liquid water content (LWC) and cloud droplet number concentration (CDNC) and exponential curves ($f(x) = a \cdot e^{-b \cdot x} + c$) for the ice crystal number concentration (ICNC), after applying cloud droplet number concentration thresholds (CDNC$_{th}$) and ice crystal number concentration thresholds (ICNC$_{th}$) on the normalized ice crystal growth rate distribution of Figs. B1 - B3. These parameters are also used to generate the curves in Fig. B1.

| Thresholds | | LWC | | CDNC | | ICNC | | |
|---|---|---|---|---|---|---|---|---|
| CDNC$_{th}$ | ICNC$_{th}$ | $m$ | r$^2$ | $m \times 10^4$ | r$^2 \times 10^3$ | $a \times 10$ | $b$ | $c$ |
| none | none | $1.15 \pm 0.02$ | 0.047 | $8.97 \pm 0.12$ | 56.04 | $3.68 \pm 0.03$ | $0.470 \pm 0.01$ | $0.814 \pm 0.003$ |
| $\geq 25$ | none | $0.65 \pm 0.03$ | 0.011 | $4.10 \pm 0.18$ | 0.872 | $4.56 \pm 0.38$ | $0.17 \pm 0.02$ | $0.701 \pm 0.039$ |
| $\geq 50$ | none | $0.51 \pm 0.03$ | 0.006 | $2.74 \pm 0.21$ | 0.337 | $5.35 \pm 0.70$ | $0.13 \pm 0.02$ | $0.625 \pm 0.072$ |
| $\geq 75$ | none | $0.43 \pm 0.04$ | 0.003 | $1.91 \pm 0.24$ | 0.141 | $6.73 \pm 1.25$ | $0.11 \pm 0.03$ | $0.494 \pm 0.127$ |
| $\geq 100$ | none | $0.46 \pm 0.04$ | 0.003 | $1.11 \pm 0.29$ | 0.041 | $6.71 \pm 1.37$ | $0.11 \pm 0.03$ | $0.498 \pm 0.139$ |
| $\geq 125$ | none | $0.48 \pm 0.05$ | 0.003 | $-0.18 \pm 0.35$ | 0.001 | $6.16 \pm 1.38$ | $0.13 \pm 0.04$ | $0.555 \pm 0.140$ |
| none | $\leq 400$ | $0.66 \pm 0.04$ | 11.96 | $3.13 \pm 0.27$ | 5.87 | $2.11 \pm 0.34$ | $44.4 \pm 9.21$ | $1.120 \pm 0.003$ |
| none | $\leq 140$ | $0.44 \pm 0.07$ | 4.80 | $2.15 \pm 0.47$ | 2.51 | $2.10 \pm 0.39$ | $46.1 \pm 13.0$ | $1.122 \pm 0.008$ |
| none | $\leq 100$ | $0.11 \pm 0.09$ | 0.30 | $1.12 \pm 0.59$ | 0.67 | $2.81 \pm 0.43$ | $18.2 \pm 8.97$ | $1.026 \pm 0.057$ |
| none | $\leq 90$ | $-0.03 \pm 0.10$ | 0.03 | $0.20 \pm 0.65$ | 0.02 | $3.27 \pm 1.15$ | $13.6 \pm 10.4$ | $0.973 \pm 0.135$ |
| none | $\leq 80$ | $-0.03 \pm 0.11$ | 0.02 | $0.28 \pm 0.71$ | 0.04 | $11.0 \pm 30.4$ | $3.28 \pm 10.5$ | $0.200 \pm 3.068$ |
| $\geq 100$ | $\leq 100$ | $-0.17 \pm 0.14$ | 0.33 | $0.23 \pm 0.81$ | 0.019 | $0.28 \pm 0.04$ | $21.1 \pm 9.79$ | $1.04 \pm 0.05$ |
| $\geq 125$ | $\leq 100$ | $-0.27 \pm 0.15$ | 0.76 | $0.13 \pm 0.87$ | 0.005 | $0.28 \pm 0.03$ | $22.8 \pm 10.1$ | $1.05 \pm 0.04$ |
| $\geq 100$ | $\leq 90$ | $-0.26 \pm 0.15$ | 0.79 | $-0.30 \pm 0.86$ | 0.032 | $0.38 \pm 0.16$ | $12.0 \pm 10.1$ | $0.93 \pm 0.18$ |
| $\geq 125$ | $\leq 90$ | $-0.37 \pm 0.16$ | 1.40 | $-0.52 \pm 0.92$ | 0.084 | $0.35 \pm 0.11$ | $14.3 \pm 10.4$ | $0.97 \pm 0.13$ |

*Author contributions.* CF conducted the scientific analysis, prepared the figures, and wrote the manuscript. CF did the reconstruction and hand-labeling of holographic data, with contributions from FR. CF, FR, AJM, NO, RS, HZ, and JH were in the field conducting the seeding experiments and measurements. KO and PS provided the vertical pointing cloud radar and one of the scanning Doppler cloud radars and maintenance of the remote sensing instrumentation. UL, JH, and FR provided supervision and scientific input during the analysis. UL, JH, and FR conceived of CLOUDLAB and obtained funding. All authors contributed to the manuscript editing and review.

*Competing interests.* The authors declare no competing interests.



*Acknowledgements.* The CLOUDLAB project has received funding from the European Research Council (ERC) under the European Union's Horizon 2020 research and innovation program (grant agreement No. 101021272 CLOUDLAB). We would like to further extend gratitude to the following people: The TROPOS PolarCAP team including Johannes Bühl, Tom Gaudek, Hannes Griesche, Willi Schimmel, and Martin Radenz for the remote sensing instrumentation and the scientific discussions and collaborations. The Meteomatics drone team, including Lukas Hammerschmidt, Daniel Schmitz, Philipp Kryenbühl, Remo Steiner, and Dominik Brändle for the support, development, and expertise of our drones. Michael Rösch (ETH) for the technical support of our field setup. Tsuneya Takahashi for providing the full data set of his ice crystal growth study. Maxime Hervo and MeteoSwiss for the wind profiler supporting our experiments. Frank Kasparek and Aleksei Shilin (Cloud Seeding Technologies) for the expertise on our seeding flares. The Swiss Army, the Gütergemeinde Hinterdorf Eriswil, and Stefan Minder for the allowing the use and maintenance of our base. To Crameri (2023) providing open source scientific color maps used in this manuscript.



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
