# Peer review of "Quantifying ice crystal growth rates in natural clouds from glaciogenic cloud seeding experiments"

_EGUsphere, 2025_

## Referee Comment (RC1)

**Review of "Quantifying ice crystal growth rates in natural clouds from glaciogenic cloud seeding experiments" by Fuchs et al.**

**Overview**

This paper examines the growth of ice crystals in mixed-phase clouds using data collected during the CLOUDLAB field campaign. CLOUDLAB is a unique experiment designed to observe the evolution of ice crystal sizes and their morphology in natural clouds. Most of the data was collected within a temperature range of -5.1°C to -8.3° C, which is associated with the columnar growth of ice particles. This study specifically investigates the growth rate of ice crystals along their major axis. The assessment of the growth rate of ice crystals was based on measurements of dimensions of single (non-aggregated) ice columns in the assumption of their linear growth. The growth rate of ice columns was calculated for three CLOUDLAB data subsets: (a) "unconstrained" diffusional growth in cloud environments restricted by certain threshold applied to LWC, CDNC, and ICNC, (b) "lucky ice crystals approach", and (c) unfiltered data. The results of this study are summarized in Fig.8, which compares ice growth rates calculated for the CLOUDLAB clouds with those from laboratory studies. The methodology utilized is robust, and the results obtained hold significant value. However, the authors restricted the results by considering the growth rates of ice crystals by their means, medians, and percentiles. It is my opinion that the most crucial aspect of this work pertains to the observation of how introduced ice particles are processed by a liquid cloud. The uniqueness of CLOUDLAB lies in its ability to observe the broadening of initially quasi-monodisperse ice particles (i.e., initial sizes limited by the sizes of frozen cloud droplets) in natural cloud environments due to varying supersaturation histories and, consequently, growth rates experienced by individual ice crystals caused by fluctuations in vertical velocity, entrainment and mixing, and recirculation through the cloud base. This information is essential for calibrating cloud models, and it would be beneficial for the completeness of this study to include this topic in the paper. Otherwise, this subject warrants a separate publication.

**Recommendation**: This study undoubtedly deserves publication in ACP. In my opinion, the novelty and importance of the presented results have passed the threshold required for acceptance for publication. I leave it to the author's discretion to decide how far they want to go in addressing my comments below.

**Comments**

1.  Section 2.5: One of the assumptions in calculating the growth rate of ice crystals ($dL/dt$) involves steady-state environmental conditions, specifically relative humidity ($RH$) and temperature ($T$), which primarily controls the ice particle growth rate. For most liquid and mixed-phase clouds, $RH$ is close to saturation over water (e.g., Korolev and Mazin, JAS, 2003; Korolev and Isaac, JAS, 2006) due to the small phase relaxation time of approximately $\tau_{ph} \sim 10^{-1} - 10^{0}$s seconds. However, $RH$ may be temporarily reduced by the entrainment of dry out-of-cloud air through the cloud top and subsequent mixing and result in the decrease of the ice growth rate. Some ice particles may encounter undersaturated environments during circulation through the cloud base which may slow down growth rate of ice particles or even result in sublimation.

Furthermore, *RH* also depends on the vertical velocity ($U_Z$) of cloud parcels. However, fluctuation $\Delta RH$ related to $U_Z$ are expected to be relatively small compared to that related to that generated by entrainment.

Temperature fluctuations ($\Delta T$) play a crucial role as well. Given that ice particles driven by turbulent fluctuations may travel between cloud top and cloud base, temperature fluctuations experienced by ice particles in boundary layer clouds will be largely determined by the cloud depth, i.e., $\Delta T = \gamma_{moist}\Delta H$.

These dynamic processes controlling fluctuations $\Delta RH$ and $\Delta T$ should be accounted in consideration of ice growth rate dL/dt. Coul you include discussion of the above effect and quantitative assessment of their contributions?

2. Entrainment and mixing are expected to be one of the main drivers of fluctuations of $\Delta RH$ and $\Delta T$. The rate of entrainment of the dry air through the cloud top depends on the intensity of turbulent fluctuations ($\varepsilon$) and depth of the cloud top inversion ($\Delta T_{inv}$). Furthermore, the effect of entrainment is most pronounced near the cloud tops of stratiform layers, and it decreases with the increase of the distance from the cloud top. Therefore, it is anticipated that in deeper stratiform clouds *dL/dt* of ice particles will be less affected by entrainment. Summarizing the above, it is anticipated that in addition to LWC, CDNC, and ICNC the growth rate of ice particles *dL/dt* will also depend on $\varepsilon$, $\Delta H$, $\Delta T_{inv}$. For the sake of completeness of the analysis it will be beneficial to the paper to include the analysis of the effect of $\varepsilon$, $\Delta H$, $\Delta T_{inv}$ of *dL/dt* similar to those as in Fig.6.

3. Section 3.2. Unfortunately, in the paper LWC, CDNC, and ICNC were introduced as major microphysical parameters affecting the growth rate of ice particles without explanation. The rational of choice of LWC, CDNC, and ICNC should be explained. It makes sense to consider adding the integral radius of cloud droplets (CDNC * mean_radius), which is reversely proportional to the time of phase relaxation $\tau_{ph}$. Such approach would be more physical compared to LWC, CDNC, since in clouds with smaller $\tau_{ph}$, *RH* is expected to be closer to saturation. It might be worth exploring potential explanation of the roll of *dL/dt* for CDNC<100cm-3 and increase of *dL/dt* with increase of LWC through the analysis of the dependence of *dL/dt* vs integral radius.

4. Figure 6a. It is worth noting that boundary layer clouds with higher LWC are expected to have higher cloud depth $\Delta H$, and therefore, ice particles may experience higher temperature fluctuations $\Delta T$. Could you explore if the increase of *dL/dt* with the increase of LWC in Fig.6a is related to the temperature fluctuations $\Delta T$ related to the cloud depth?

5. During columns' growth, their hollowness depends on *RH*, i.e., when *RHice* is close to its saturation over ice, then columns grow as solid hexagonal prisms. In contrast, when *RHice* increases and approaches saturation over liquid, the columns develop hollowness. As was shown by Harrington et al. (https://ams.confex.com/ams/105ANNUAL/meetingapp.cgi/Paper/455629, video recording of the presentation is available from the AMS site), the rate of changes of the hollowness is a function of *RH*. In other words, if the images of columns show varying rates of changing of the hollowness along the c-axis, then it is indicative that the ambient *RH(t)* related to this

specific ice crystal was not constant. The analysis (qualitative or quantitative) of the patterns of hollows developed inside columns may shed light on the growth condition of studied "lucky" ice particles. The authors may consider adding this consideration to the paper.

6. Page 10, Line 225 "*The riming efficiency of ice crystals strongly depends on the ice crystal size (Wang and Ji, 2000) and CDNC and is highest for large ice crystals and high CDNC.*" The riming rate also depends on the droplet size (Wang and Ji, 2000). The above statement should be modified to include the dependence on the droplet size.

7. Page 17, Line 375 "*The temperature dependence of our growth rates shows reasonable agreement with laboratory studies, although the absolute values tend to be lower for our in situ observations. This discrepancy likely arises from differences in growth conditions: laboratory setups typically provide unlimited water vapor, high LWC, and isolated ice crystal growth, whereas our in situ observations involve limited water vapor availability and high ICNC*" It would be more accurate to rewrite this statement in terms of fluctuations $\Delta RH$ and $\Delta T$, rather that "limited" and "unlimited" LWC. Thus, in laboratory studies usually $RH$ and $T$ are maintained to remain constant with relatively small fluctuations $\Delta RH$ and $\Delta T$. Whereas, in natural clouds fluctuations $\Delta RH$ and $\Delta T$ may be significantly higher compared to those in lab studies due to the effects of $\varepsilon$, $\Delta H$, $\Delta T_{inv}$, $U_z$, LWC, CDNC, and ICNC.

8. It is worth adding a discussion that the frequency distribution of $RH$ in stratiform mixed-phase clouds is skewed towards smaller values (e.g., Korolev and Isaac, 2006) primarily due to entrainment and the WBF process. Therefore, the net effect of such skewness will be lower ice growth rates $dL/dt$ compared to the lab studies.

9. In my opinion, the results in Fig.6 are crucial for parameterizing ice growth rates in stratiform clouds and improving cloud models. Laboratory studies measure $dL/dt$ under constant $RH$ and $T$, but CLOUDLAB includes effects of turbulence, radiation transfer, riming, and aggregation, which can't be fully replicated in the lab. The obtained dependencies of $dL/dt$ vs LWC, CDNC, and ICNC are a very valuable outcome of this study, and I highly recommend expanding consideration and discussion around this question.

10.     Figure 7(g,h,i). What are the black solid lines at normalized_growth_rate =1?

Alexei Korolev

---

## Author Comment (AC1)

**Response to Referees**
**Quantifying ice crystal growth rates in natural clouds from glaciogenic cloud seeding experiments**

Christopher Fuchs, Fabiola Ramelli, Anna J. Miller, Nadja Omanovic, Robert Spirig,
Huiying Zhang, Patric Seifert, Kevin Ohneiser, Ulrike Lohmann, and Jan Henneberger

July 4, 2025

Dear editor and referees,

We sincerely thank Luis A. Ladino as editor for handling our manuscript and Alexei Korolev and Darrel Baumgardner for their thorough and insightful reviews. Below, we present detailed point-by-point responses to all comments. Reviewer comments are shown in purple boxes, our responses are provided in plain text, and modifications to the manuscript are indicated in italic font. We have also uploaded a version of the manuscript with highlighted track changes, which includes both major revisions and minor typesetting corrections.

Best regards,
Christopher Fuchs et al.

**Response to reviewer comment #1 by Alexei Korolev**

**Comments to the Author**

This paper examines the growth of ice crystals in mixed-phase clouds using data collected during the CLOUDLAB field campaign. CLOUDLAB is a unique experiment designed to observe the evolution of ice crystal sizes and their morphology in natural clouds. Most of the data was collected within a temperature range of -5.1 °C to -8.3 °C, which is associated with the columnar growth of ice particles. This study specifically investigates the growth rate of ice crystals along their major axis. The assessment of the growth rate of ice crystals was based on measurements of dimensions of single (non-aggregated) ice columns in the assumption of their linear growth. The growth rate of ice columns was calculated for three CLOUDLAB data subsets: (a) "unconstrained" diffusional growth in cloud environments restricted by certain threshold applied to LWC, CDNC, and ICNC, (b) "lucky ice crystals approach", and (c) unfiltered data. The results of this study are summarized in Fig.8, which compares ice growth rates calculated for the CLOUDLAB clouds with those from laboratory studies. The methodology utilized is robust, and the results obtained hold significant value. However, the authors restricted the results by considering the growth rates of ice crystals by their means, medians, and percentiles. It is my opinion that the most crucial aspect of this work pertains to the observation of how introduced ice particles are processed by a liquid cloud. The uniqueness of CLOUDLAB lies in its ability to observe the broadening of initially quasi-monodisperse ice particles (i.e., initial sizes limited by the sizes of frozen cloud droplets) in natural cloud environments due to varying supersaturation histories and, consequently, growth rates experienced by individual ice crystals caused by fluctuations in vertical velocity, entrainment and mixing, and recirculation through the

cloud base. This information is essential for calibrating cloud models, and it would be beneficial for the completeness of this study to include this topic in the paper. Otherwise, this subject warrants a separate publication.

**Recommendation:** This study undoubtedly deserves publication in ACP. In my opinion, the novelty and importance of the presented results have passed the threshold required for acceptance for publication. I leave it to the author's discretion to decide how far they want to go in addressing my comments below.

We sincerely thank Alexei Korolev for the thorough and insightful review of our manuscript and for recognizing the novelty and significance of our work. We have carefully considered all of the points raised and have revised and extended the manuscript accordingly.

**Comments**

**1.** Section 2.5: One of the assumptions in calculating the growth rate of ice crystals ($dL/dt$) involves steady-state environmental conditions, specifically relative humidity ($RH$) and temperature ($T$), which primarily controls the ice particle growth rate. For most liquid and mixed-phase clouds, $RH$ is close to saturation over water (e.g., Korolev and Mazin, JAS, 2003; Korolev and Isaac, JAS, 2006) due to the small phase relaxation time of approximately $\tau_{ph} \sim 10^{-1} - 10^0$ seconds. However, $RH$ may be temporarily reduced by the entrainment of dry out-of-cloud air through the cloud top and subsequent mixing and result in the decrease of the ice growth rate. Some ice particles may encounter undersaturated environments during circulation through the cloud base which may slow down growth rate of ice particles or even result in sublimation.
Furthermore, $RH$ also depends on the vertical velocity ($U_Z$) of cloud parcels. However, fluctuation $\Delta RH$ related to $U_Z$ are expected to be relatively small compared to that related to that generated by entrainment.
Temperature fluctuations ($\Delta T$) play a crucial role as well. Given that ice particles driven by turbulent fluctuations may travel between cloud top and cloud base, temperature fluctuations experienced by ice particles in boundary layer clouds will be largely determined by the cloud depth, i.e., $\Delta T = \gamma_{moist}\Delta H$. These dynamic processes controlling fluctuations $\Delta RH$ and $\Delta T$ should be accounted in consideration of ice growth rate $dL/dt$. Could you include discussion of the above effect and quantitative assessment of their contributions?

Yes, you are absolutely right, ice crystal growth is primarily controlled by the history of $RH$ (or supersaturation with respect to ice) and temperature that the ice crystals experienced since nucleation. These conditions are likely subject to fluctuations caused by processes such as entrainment of dry air, recirculation through the cloud base, or up- and downdrafts.
Unfortunately, we do neither have direct measurements of $\Delta RH$ nor $\Delta T$ that would allow for a quantitative assessment of their influence. This is a common challenge when conducting in situ measurements in supercooled clouds: sensor icing and persistent conditions near $RH \approx 100\,\%$ complicate accurate measurements of $RH$ and temperature fluctuations using most commercially available sensors.
Therefore, we are still required to assume a steady-state environment for our experiments. We expect that the magnitude of $RH$ and temperature fluctuations do not vary drastically between the different days or clouds sampled in this study, that allows for comparability on different days. The targeted low stratus clouds are relatively unique in their dynamical properties, exhibiting fewer and weaker dynamical features compared to most other cloud types.
Notably, our lucky ice crystal filtering approach likely provides the best representation of ideal growth conditions under sustained high supersaturation. By selecting the largest crystals, presumably those that experienced consistently favorable growth conditions (highest supersaturation), this method may partially account for the effects of $\Delta RH$ and $\Delta T$ variability along the growth trajectory. It also allows us to assess the combined relative impact of these fluctuations by comparing the results to those obtained from the

unfiltered data set.

We plan to address these open questions more thoroughly in our follow-up project, scheduled to begin in December 2025, where we aim to utilize a quasi-Lagrangian measurement approach using a free-floating balloon that follows the seeding plume. However, even with this approach, the sensors are expected to face challenges due to the previously mentioned measurement conditions.

Nevertheless, we include this discussion to highlight the current limitations of the CLOUDLAB measurements and to emphasize where future improvements are needed, particularly with respect to unconstrained variables and unresolved influences on ice crystal growth.

We adapted and added sentences in lines 326 - 329: *"This filtering strategy may offer some insight into the influence of unresolved fluctuations in temperature and supersaturation along the ice crystal trajectory. By selecting the largest, "lucky", ice crystals, i.e., presumably those that experienced consistently favorable conditions such as sustained supersaturation, the method provides a way of indirectly assessing the impact of environmental variability on ice crystal growth, which is otherwise not directly assessable in our data set."*

We adapted and added sentences in lines 403-410: *"In contrast, our in situ observations reflect natural cloud environments, which are subject to variability in temperature, relative humidity, and supersaturation with respect to ice indicated by the spread in our data set. These fluctuations, particularly in supersaturation, are likely to reduce the rates of ice crystal growth. Supersaturation in clouds is typically capped just above water saturation due to the high CDNC, resulting in fastest ice crystal growth. However, during our seeding experiments, it can extend to significantly lower values by approaching ice saturation within fully glaciated pockets. Further, the WBF process itself and entrainment of dry air can skew the relative humidity distribution toward lower values (Korolev and Isaac, 2006). These effects result in a net reduction in vapor diffusional growth rates of ice crystals, leading to suppressed growth compared to idealized laboratory conditions."*

We adapted and added sentences in lines 415-417: *"While not a direct measurement of environmental variability, this approach offers a way to partially assess the influence of temperature and supersaturation fluctuations by isolating ice crystals that presumably grew under persistently favorable conditions."*

**2.** Entrainment and mixing are expected to be one of the main drivers of fluctuations of $\Delta RH$ and $\Delta T$. The rate of entrainment of the dry air through the cloud top depends on the intensity of turbulent fluctuations ($\Delta RH$) and depth of the cloud top inversion ($\Delta T_{\text{inv}}$). Furthermore, the effect of entrainment is most pronounced near the cloud tops of stratiform layers, and it decreases with the increase of the distance from the cloud top. Therefore, it is anticipated that in deeper stratiform clouds $dL/dt$ of ice particles will be less affected by entrainment. Summarizing the above, it is anticipated that in addition to LWC, CDNC, and ICNC the growth rate of ice particles $dL/dt$ will also depend on $\varepsilon$, $\Delta H$, $\Delta T_{\text{inv}}$. For the sake of completeness of the analysis it will be beneficial to the paper to include the analysis of the effect of $\varepsilon$, $\Delta H$, $\Delta T_{\text{inv}}$ of $dL/dt$ similar to those as in Fig.6.

We agree that entrainment and mixing are important drivers of fluctuations in $\Delta RH$ and $\Delta T$, which are closely linked to turbulence ($\varepsilon$), cloud depth ($\Delta H$), and inversion strength ($\Delta T_{\text{inv}}$), and thereby also affect $dL/dt$. However, our dataset does not allow for a detailed evaluation of these influences in a way comparable to Fig. 6. The data presented in Fig. 6 are fully co-located, derived from HOLIMO within the same measurement volume, and time-resolved throughout each seeding experiment, enabling to quantify the interconnected variability of the variables shown in the figure.

In contrast, the parameters $\varepsilon$, $\Delta H$, and $\Delta T_{\text{inv}}$ would only be available only as single values per seeding experiment, derived from remote sensing or radiosonde profiles. These measurements are based on larger

sampling volumes and different locations, as the balloon with in situ instruments is usually located further down wind of the radar observed volume. As a result, we would be limited to comparing mean values of $dL/dt$ across experiments. However, these mean values are mainly controlled by the temperature variation between experiments. Given the limited number of seeding experiments (14 in total) conducted at varying temperatures, a normalization similar to that used in Fig. 6 is unfortunately not feasible.

A more detailed assessment of these dynamical influences would require fully co-located and time-resolved observations, which we aim to explore in future work using a free floating balloon system following the seeding patch.

> **3.** Section 3.2. Unfortunately, in the paper LWC, CDNC, and ICNC were introduced as major microphysical parameters affecting the growth rate of ice particles without explanation. The rational of choice of LWC, CDNC, and ICNC should be explained. It makes sense to consider adding the integral radius of cloud droplets (CDNC * mean_radius), which is reversely proportional to the time of phase relaxation $\tau_{ph}$. Such approach would be more physical compared to LWC, CDNC, since in clouds with smaller $\tau_{ph}$, $RH$ is expected to be closer to saturation. It might be worth exploring potential explanation of the roll of $dL/dt$ for CDNC $< 100\,\mathrm{cm}^{-3}$ and increase of $dL/dt$ with increase of LWC through the analysis of the dependence of $dL/dt$ vs integral radius.

Thank you for this remark. While LWC, CDNC, and ICNC are common primary and key microphysical parameters we used them as starting point for our analysis. However, with respect to ice crystal growth, you are absolutely right that the integral radius of cloud droplets is a physically more meaningful parameter than LWC and CDNC. We have therefore extended our analysis to include the integral radius as an additional variable, as shown below. This analysis, however, revealed that incorporating the integral radius has no significant effect on our results and does not alter the main findings of the study. Therefore, we decided not to include the integral radius as a primary variable in the manuscript, given the broader relevance and comparability of LWC, CDNC, and ICNC to other studies in both observational and modeling contexts.

[Figure]

Figure R1: Vapor diffusional ice crystal growth rates for the 14 seeding experiments (i.e., SE 1) versus liquid water content (LWC) in **(a)**, cloud droplet number concentration (CDNC) in **(b)**, mean cloud droplet diameter in **(c)**, and integral radius in **(d)**. Vapor diffusional growth rates are shown as mean values of the respective bins and are plotted at the bin centers with bin sizes of $0.05\,\mathrm{g\,m}^{-3}$ for the LWC, $80\,\mathrm{cm}^{-3}$ for the CDNC, $2\,\mu\mathrm{m}$ for the droplet diameter, and $800\,\mu\mathrm{m\,cm}^{-3}$. Values are shown only if $> 10$ ice crystals fall in the bin, and unfilled markers indicate bins with 10 - 25 ice crystals. Experiments with weak or no riming are shown as solid lines and experiments with strong riming are shown as dotted lines. The color indicates the temperature of each seeding experiment.

In Fig. R1, which is structured similarly to Fig. 4, panels c and d have been modified: the ICNC has been replaced by the mean droplet diameter in panel Fig. R1c, and by the integral radius of cloud droplets in panel Fig. R1d. The growth rates versus mean droplet diameter (see Fig. R1c) shows a decrease with increasing diameter. The figure also indicates that the mean cloud droplet diameter typically ranges between 10 and 15 $\mu$m. The relationship between ice crystal growth rates and the integral droplet radius (see Fig. R1d) reveals a general increase in growth rates with larger integral radii. Qualitatively, these trends show that the observed variations in growth rates are primarily driven by CDNC, with only a minor influence from cloud droplet size.

[Figure]

Figure R2: Normalized ice crystal growth rates distributions (weak riming seeding experiments only see Table 1), vs LWC in the (1st column), CDNC in the (2nd column), droplet mean diameter (3rd column), integral radius for cloud droplet (4th column), and ICNC (5th column). The mean growth rates (white circles) and respective $\pm 1\sigma$ standard deviations (solid black lines) for each column of bins are shown to better highlight trends of the distributions. Linear fits are used for the LWC, CDNC, and integral radius, and exponential fits for the ICNC (solid orange; 95% confidence interval shading). If threshold are applied they are given in the lower left corner of the respective panel. 1st row **(a)-(e)**: no thresholds are applied. 2nd row **(f)-(j)**: CDNC $\geq 100\,\mathrm{cm}^{-3}$. 3rd row **(k)-(o)**: integral radius $\geq 500\,\mu\mathrm{m\,cm}^{-3}$. 3th row **(p)-(t)**: CDNC $\geq 100\,\mathrm{cm}^{-3}$, integral radius $\geq 100\,\mu\mathrm{m\,cm}^{-3}$, and ICNC $\leq 100\,\mathrm{cm}^{-3}$.

Figure R2 presents the normalized growth rates versus LWC, CDNC, cloud droplet mean diameter, integral radius, and ICNC, showing similar patterns to those observed in Fig. R1. For the unfiltered dataset (first row of Fig. R2), growth rates decrease with increasing cloud droplet mean diameter, with most values concentrated between 10 and $15\,\mu\mathrm{m}$ (see Fig. R2c). The dependence of growth rate on the integral radius (Fig. R2d) closely resembles the trends observed for CDNC (Fig. R2b), with growth rates increasing up to an integral radius of approximately $500\,\mu\mathrm{m}\,\mathrm{cm}^{-3}$, beyond which they level off. This plateau behavior suggests that, similar to CDNC, the integral radius also provides a meaningful threshold ($\geq 500\,\mu\mathrm{m}\,\mathrm{cm}^{-3}$) approximating unconstrained ice crystal growth conditions.

We now evaluate how applying the CDNC threshold (second row of Fig. R2) and the integral radius threshold (third row) affects the growth rate distributions, and compare the resulting changes. Surprisingly, the CNDC and integral radius threshold yield nearly identical impacts on the growth rate distributions if applied individually. The variations observed in the corresponding fit lines are statistically insignificant. In the fourth row of Fig. R2, we apply the combined thresholds used in the main manuscript, $\mathrm{CDNC} \geq 100\,\mathrm{cm}^{-3}$ and $\mathrm{ICNC} \leq 100\,\mathrm{cm}^{-3}$, together with the newly tested integral radius threshold ($\geq 100\,\mu\mathrm{m}\,\mathrm{cm}^{-3}$). The resulting growth rate distributions show no substantial differences compared to the second row of Fig. B3 (final thresholds used for our analysis), and the corresponding fit parameters also exhibit no significant deviations.

Based on these results, and the absence of significant changes after introducing the integral radius as a new variable, we conclude that our original methodology is robust, and there is no compelling need to include the integral radius as an additional key parameter.

We added new sentences in lines 255-265: *"The choice of LWC, CDNC, and ICNC as the primary microphysical variables in our analysis is motivated by their widespread use in both observational and modeling studies, as well as their direct relevance for characterizing clouds. While these quantities are commonly reported and easy to interpret, we acknowledge that the integral radius of cloud droplets, defined as the product of CDNC and mean droplet radius, also serves as a representative microphysical property relevant to the vapor diffusional growth of ice crystals. Under conditions with a high integral radius, the available surface area of cloud droplets is larger, allowing the water vapor field to be more effectively replenished through cloud droplet evaporation. We tested the integral radius as an additional variable in our analysis, showing that it exhibits trends very similar to CDNC and appears to be strongly dominated by it. Therefore, we decided not to include the integral radius as a primary variable. This also supports the robustness of our variable selection in capturing the dominant microphysical influences on vapor diffusional ice crystal growth. Note: The seeding experiments with strong riming (see Sect. 3.1 and Table A1) are excluded from this analysis."*

**4.** Figure 6a. It is worth noting that boundary layer clouds with higher LWC are expected to have higher cloud depth $\Delta H$, and therefore, ice particles may experience higher temperature fluctuations $\Delta T$. Could you explore if the increase of $dL/dt$ with the increase of LWC in Fig.6a is related to the temperature fluctuations $\Delta T$ related to the cloud depth?

We agree that cloud depth ($\Delta H$) can modulate the temperature fluctuations ($\Delta T$) that ice crystals experience and may therefore influence their growth rates. This effect could plausibly contribute to the relationship between $dL/dt$ and LWC shown in Fig. 6a, given that higher LWC values are often associated with deeper clouds. However, as with the parameters discussed in Comment 2, our dataset does not support a time-resolved or co-located analysis of this effect. Estimates of $\Delta H$ are only available as single values per seeding experiment, retrieved from remote sensing, and cannot be linked directly to the in situ measurements of ice growth at the particle level.

Because the variation in $dL/dt$ between seeding experiments is strongly driven by temperature and we only have 14 seeding experiments available conducted across a temperature range, we are unable to normalize

for this temperature dependence and isolate a potential influence of $\Delta H$ and linkage to the LWC within this framework.

**5.** During columns' growth, their hollowness depends on $RH$, i.e., when $RHice$ is close to its saturation over ice, then columns grow as solid hexagonal prisms. In contrast, when $RHice$ increases and approaches saturation over liquid, the columns develop hollowness. As was shown by Harrington et al. (`https://ams.confex.com/ams/105ANNUAL/meetingapp.cgi/Paper/455629`, video recording of the presentation is available from the AMS site), the rate of changes of the hollowness is a function of $RH$. In other words, if the images of columns show varying rates of changing of the hollowness along the c-axis, then it is indicative that the ambient $RH(t)$ related to this specific ice crystal was not constant. The analysis (qualitative or quantitative) of the patterns of hollows developed inside columns may shed light on the growth condition of studied "lucky" ice particles. The authors may consider adding this consideration to the paper.

That is an excellent idea, to link variations in the hollowness of observed ice crystals to their history of experienced relative humidity. However, pursuing this analysis, even for a qualitative assessment, would be highly time-consuming and is therefore beyond the scope of the current study. Nevertheless, we consider this a promising research direction and have decided to advertise it as a Master's thesis project for the upcoming semester.

**6.** Page 10, Line 225 *"The riming efficiency of ice crystals strongly depends on the ice crystal size (Wang and Ji, 2000) and CDNC and is highest for large ice crystals and high CDNC."* The riming rate also depends on the droplet size (Wang and Ji, 2000). The above statement should be modified to include the dependence on the droplet size.

Yes, indeed! The modified sentence in line 233-234 now reads: *"The riming rate of ice crystals strongly increases with increasing ice crystal size, cloud droplet size, and CDNC (Wang and Ji, 2000)."*

**7.** Page 17, Line 375 *"The temperature dependence of our growth rates shows reasonable agreement with laboratory studies, although the absolute values tend to be lower for our in situ observations. This discrepancy likely arises from differences in growth conditions: laboratory setups typically provide un-limited water vapor, high LWC, and isolated ice crystal growth, whereas our in situ observations involve limited water vapor availability and high ICNC"* It would be more accurate to rewrite this statement in terms of fluctuations $\Delta RH$ and $\Delta T$, rather that "limited" and "unlimited" LWC. Thus, in laboratory studies usually $RH$ and $T$ are maintained to remain constant with relatively small fluctuations $\Delta RH$ and $\Delta T$. Whereas, in natural clouds fluctuations $\Delta RH$ and $\Delta T$ may be significantly higher compared to those in lab studies due to the effects of $\varepsilon$, $\Delta H$, $\Delta T_{\text{inv}}$, $U_Z$ LWC, CDNC, and ICNC.

Yes, this is correct. We have revised our statement to more accurately reflect the physical drivers of vapor diffusional ice crystal growth, namely temperature and supersaturation with respect to ice.

The new statement in lines 398-402 reads: *"The temperature dependence of our observed ice crystal growth rates shows reasonable agreement with laboratory studies, although the absolute values tend to be lower in our in situ observations. This discrepancy likely arises from differences in growth conditions. Laboratory experiments typically maintain steady-state environments with constant temperature and relative humidity, providing an effectively unlimited supply of water vapor. Moreover, they often examine the growth of isolated ice crystals in the absence of competition for water vapor."*

**8.** It is worth adding a discussion that the frequency distribution of $RH$ in stratiform mixed-phase

> clouds is skewed towards smaller values (e.g., Korolev and Isaac, 2006) primarily due to entrainment and the WBF process. Therefore, the net effect of such skewness will be lower ice growth rates $dL/dt$ compared to the lab studies.

Thank you for pointing out this feature. It provides an additional explanation for why ice crystal growth rates observed in natural clouds tend to be lower than those measured in laboratory conditions.

We have adapted and extended the sentences in lines 407-408: *"Further, the WBF process itself and entrainment of dry air can skew the relative humidity distribution toward lower values (Korolev and Isaac, 2006)."*

> **9.** In my opinion, the results in Fig.6 are crucial for parameterizing ice growth rates in stratiform clouds and improving cloud models. Laboratory studies measure $dL/dt$ under constant RH and $T$, but CLOUDLAB includes effects of turbulence, radiation transfer, riming, and aggregation, which can't be fully replicated in the lab. The obtained dependencies of $dL/dt$ vs LWC, CDNC, and ICNC are a very valuable outcome of this study, and I highly recommend expanding consideration and discussion around this question.

We thank for this encouraging comment. We agree that the relationships shown in Fig. 6 provide a valuable observational basis for improving microphysical parameterizations of ice crystal growth in stratiform clouds. In response to your suggestion, we have expanded the discussion to highlight the relevance of these dependencies, particularly their potential to inform and constrain ice growth representations in models.

We included a new paragraph in lines 391-395: *"Nonetheless, the full (unfiltered) dataset and its relationships between ice crystal growth rates and LWC, CDNC, and ICNC provides valuable observational constraints for cloud microphysical parameterizations. Unlike laboratory data, our measurements reflect the integrated effects of ice crystal growth under variable and natural mixed-phase cloud environments. As such, they can help inform and refine parameterizations in numerical models that aim to realistically capture ice growth processes in stratiform cloud systems (e.g., Omanovic et al., 2025)."*

> **10.** Figure 7(g,h,i). What are the black solid lines at normalized_growth_rate =1?

Unfortunately, there may have been an issue with figure rendering or display, as there should not be a horizontal black line at normalized growth rate = 1 in Fig. 7g, h, or i. We are not aware of any such line being present in the original figure and are unsure why it may appear in your version.
For clarification, the black lines shown in these panels represent the fitted growth rate trends for the subset of data filtered by CDNC $> 100\,\mathrm{cm}^{-3}$ and ICNC $< 100\,\mathrm{L}^{-1}$, along with their 95 % confidence intervals, as indicated in the legend.
The black horizontal bars at the bottom of Fig. 7h and i indicate the range of growth rate data included after applying the respective thresholds, similar to the visualization in Fig. 7b and f.

**Response to reviewer comment #2 by Darrel Baumgardner**

**Comments to the Author**

> The work presented here is an elegant approach to validating laboratory crystal growth studies using in situ measurements. I think that the authors have done a commendable job of putting the results into an understandable and, mostly, defendable arguments. I say "mostly" because I do have a number of concerns that I need to be addressed before I accept this for publication. Even though I gave the paper "excellents" in the three categories, there are a number of issues that need clarification. In addition, as I will state in my concluding remarks, because of the implications that this study has for operational cloud seeding, I strongly suggest that the authors include in the introduction and concluding remarks some discussion on how the results from this study can improve the current state of glaciogenic seeding.

We sincerely appreciate your positive feedback on the quality and significance of our manuscript. While the primary focus of this study is to advance the fundamental understanding of cloud microphysical processes rather than operational cloud seeding applications, we agree that the implications for glaciogenic seeding are important. In response to your suggestion, we will include revisions to better highlight how our findings could contribute to improving cloud seeding operations, while maintaining the emphasis on fundamental research. We would also like to point out that complementary studies by Chen et al. (2024) and Miller et al. (2025) have specifically addressed many aspects critical to glaciogenic cloud seeding, including the ice-nucleating properties of seeding particles. To avoid overlap with these works and to retain a clear focus, we aim to incorporate the implications on cloud seeding concisely without shifting the main scope of the present paper.

**Issues:**

> **1)** I have not read the Ramelli et al (2024) papers, so if this question was addressed there, then it should be reiterated in this paper because of its relevance to connecting what is produced by the burning flare to what is measured downwind. The flare burns for 5-6 minutes and contains approximately 200 g of material, 20 g of which is silver iodide. Two questions: 1) How many ice crystals would be expected to activate from 20 g of AgI and are the number of new crystals detected downwind consistent with expectations? and 2) What is the composition of the other 180 g of material and is it completely hydrophobic, i.e. no possibility of being IN or CCN?

Yes, these are indeed two highly relevant and important questions! Unfortunately, they are also very difficult to answer quantitatively, although we are equally interested in more definitive answers. Before going into more detail, we would like to highlight two recent studies that address both questions in high detail:

First, the study by Chen et al. (2024), "Critical Size of Silver Iodide Containing Glaciogenic Cloud Seeding Particles", investigates the ice nucleating ability of aerosols generated by flares similar in composition to ours, though somewhat smaller in size. It specifically examines how the seeding aerosol size affects their efficiency as ice nucleating particles under mixed-phase cloud conditions.

Second, the work by Miller et al. (2025), "Quantified Ice-Nucleating Ability of AgI-Containing Seeding Particles in Natural Clouds", (a co-author of this manuscript), tries to answer particularly the first question. However, despite the extensive work a full quantitative answer could not be given.

Since we cannot yet provide a complete answer to both of your questions, we offer instead a brief summary of the current state of our knowledge.

Regarding question 1): The main difficulty in estimating how many ice crystals are produced from the 20 g of AgI lies in the lack of detailed knowledge about the size distribution and exact composition of the aerosols after the flare burning. However, we expect that the majority of the generated aerosols

do contain AgI. Furthermore, we have developed a hypothesis for the freezing mechanism and consider contact freezing not being a dominant pathway, because we do not see a correlation between CDNC and ice-nucleated fraction (INF).

Miller et al. (2025) showed that there is a strong linear correlation between the ICNC and seeding aerosol concentration. The inferred ice-nucleated fractions are between $0.1\%$ and $1\%$ (median values) showing a slight increase with decreasing ambient temperature. Further, regarding the freezing mechanism Miller et al. (2025) [Atmos. Chem. Phys., 25, page 5399] states:

> "We expect that the polydisperse seeding particles grow hygroscopically upon emission into the cloud at water saturation, and/or some particles may activate into cloud droplets if encountering local regions of supersaturation; then the best particles (largest and/or most quickly reaching aw≈1) initiate freezing. We do not rule out the possibility that contact freezing may also occur, but we do not consider it to be dominant due to the fact that there were no positive correlations found between INFs and background CDNC or between INFs and residence time, and because the collision rates are expected to be very low (particle sizes are in the Greenfield gap). Furthermore, we also suggest that ice nucleation occurs mostly at the start of the experiment, because once there is ice, the reduced saturation ratio causes shrinking/evaporation of leftover droplets, inducing freezing point depression and reducing the likelihood of further nucleation. These effects of more limited water vapor may explain why our observed INFs were an order of magnitude lower than the recent measurements of the same seeding flare particles by Chen et al. (2024)."

In addition, Chen et al. (2024) investigated the ice-nucleating ability of aerosols generated from similar seeding flares under laboratory conditions. They found that particles larger than approximately $90\,\text{nm}$ exhibit ice nucleation efficiencies comparable to pure AgI particles, whereas smaller particles show reduced efficiency, likely due to the presence of non-AgI components. Importantly, their study also highlighted that the size distribution of flare-generated aerosols is sensitive to the ambient wind speed during combustion: higher wind speeds form smaller particles, which potentially shifts the size distribution toward less favorable sizes for ice nucleation. However, under typical operational conditions, the majority of particles are expected to exceed the critical $90\,\text{nm}$ threshold.

Regarding question 2): Unfortunately, the exact composition of the flares is proprietary and therefore not fully disclosed. However, it is known that the Zeus MK2 flares (Cloud Seeding Technologies, Germany) contain $200\,\text{g}$ of material, comprising $11.8\%$ AgI, $15.3\%$ iodine-containing compounds, and additional components such as ammonium perchlorate, catalysts, and a fuel binder (Chen et al., 2024). The exact chemical composition of the aerosols after combustion is even less well characterized.

Regarding the INP and CCN abilities, Chen et al. (2024) [Geophysical Research Letters, 51, e2023GL106680, page 6] states:

> "The flare-generated particles contain other soluble components to promote droplet activation before freezing in real-world cloud-seeding operations. In particular, non-AgI components like ammonium perchlorate, and volatile compounds were produced upon burning."

Thus, it is expected that some fraction of the emitted aerosols are hygroscopic and can act as CCN. The study by Miller et al. (2025) further supports this, noting (see quotation from question 1) that the seeding particles first grow hygroscopically upon entering the cloud environment, consistent with CCN behavior. Regarding the ice-nucleation (IN) activity of non-AgI materials: we did not find explicit evidence that these are ice active. Instead, their presence appears to dilute the AgI content, leading to a reduction in IN efficiency, particularly in the smaller particle size range.

In our follow-up project, scheduled to begin in December 2025, an entire work package will be dedicated to hygroscopic seeding. There, we will continue to use the same glaciogenic cloud seeding flares,

among others, for hygroscopic seeding to better quantify the seeding particles ability to act as CCN.

We adapted and added references in lines 65-73: *A customized uncrewed aerial vehicle (UAV, Meteo-drone MM-670, Meteomatics AG, Switzerland) (see Miller et al., 2024) is equipped with a burn-in-place flare (Zeus MK2, Cloud Seeding Technologies, Germany) containing approximately 200 g of seeding material. The flare composition includes around 20 g of silver iodide (AgI) along with other compounds that are ice-active at temperatures below -5°C (Chen et al., 2024). The exact composition of the flares is proprietary and not fully disclosed; moreover, the chemical composition of the seeding particles after combustion is even less well characterized. Chen et al. (2024) showed that seeding particles larger than approximately 90 nm exhibit ice nucleation efficiencies comparable to pure AgI particles, while smaller particles show reduced efficiency, likely due to the presence of non-AgI components. Additionally, Miller et al. (2025) found that the number of ice crystals produced from the seeding particles is linearly correlated with the seeding particle concentration and showed ice-nucleated fractions between 0.1 % and 1 %.*

> **2)** Nothing is mentioned about the water vapor pressures in these clouds, i.e. even if ice crystals are activated in supersaturation (SS) wrt. ice $< 100\%$, they can't grow if the environment is not $> 100\%$ SS wrt ice. What is maintaining these clouds SS, just the WBF mechanism?

Thank you for raising this important point. Indeed, ice crystals will only grow if the ambient environment is supersaturated with respect to ice. Initially, the supersaturation with respect to ice is primarily maintained by the WBF mechanism. After complete glaciation, when the liquid water content (LWC) reaches zero, the ice crystals continue to grow until a new equilibrium is established at ice saturation.

The persistence of low stratus clouds suggests the presence of a source of supersaturation. These clouds predominantly form in winter (November–February) under synoptic-scale northeasterly winds within the Swiss Plateau, a confined basin bordered by the Jura Mountains to the north and the Alps to the south, and are capped by a strong inversion layer. Their persistence, and the sustained supersaturation, are likely driven by (i) the low incoming of solar radiation combined with radiative cooling at the cloud top, and (ii) a continuous influx of moist air (synoptically and from Swiss lakes and rivers) encountering orographic blocking and turbulence within the boundary layer which both promotes the lifting of moist air. Their persistency and formation however is a highly complex process which is yet now well constrained (Scherrer and Appenzeller, 2014; Westerhuis et al., 2020).

> **3)** The CDNCs of up to 400-500 cm$^{-3}$ seem quite high for a wintertime, shallow stratiform cloud, can the authors reference other studies in such clouds where the CDNCs are this high?

Thank you for raising this concern! Indeed, CDNC of 400–500 cm$^{-3}$ are on the higher side for stratiform clouds, but they are not unprecedented. Within our own dataset, the exemplary seeding experiment shown in Fig. 2 represents one of the higher cases, with CDNCs across all experiments ranging between 170 and 540 cm$^{-3}$. Unfortunately, to our knowledge, there are no other published references reporting wintertime CDNC measurements in low stratus clouds over Switzerland for direct comparison. However, the compilation by Miles et al. (2000) shows that continental stratiform clouds can exhibit CDNC values reaching almost 700 cm$^{-3}$. Thus, while our values are on the higher end, they are still within the range observed elsewhere.

It is also noteworthy that low stratus clouds, especially in winter, are closely linked to fog formation and are sometimes referred to as high fog. Fog droplet concentrations can reach up to 600 cm$^{-3}$ under certain conditions (Pruppacher and Klett, 2010), providing additional context for the high CDNCs we observe.

Finally, it is important to emphasize that the sampled low stratus clouds formed under polluted continental conditions. The Swiss Plateau, where the measurements were taken, is confined by the Jura Mountains to the north and the Alps to the south, creating a topographic basin. In wintertime, anthropogenic emissions, coupled with reduced lateral and vertical mixing under stable stratification, lead to elevated CCN

concentrations. For example, Schmale et al. (2018) observed CCN concentrations exceeding $1000\,\mathrm{cm}^{-3}$ at a rural European site (Melpitz, Germany) during winter.

We included two sentences in lines 116-118: *While the observed CDNC is relatively high, it is not unprecedented for continental stratiform clouds (Miles et al., 2000). This particular example lies at the upper end of the range observed in our dataset, which spans CDNC values between 170 and $540\,\mathrm{cm}^{-3}$.*

**Ramifications for operational glaciogenic seeding**

Cloud seeding to enhance precipitation remains a controversial topic, particularly since many cloud seeding operators are inclined to use their "instincts" when dispersing seeding material, rather than taking a more scientific approach, i.e. selecting the time and location to release material based on cloud microphysical conditions. The results of this study have clear implications with respect to assisting seeding programs to improve their effectiveness. I think a word to that effect in the introduction then a follow-up in the conclusions would be a help to the weather modification community.

Thank you for this suggestion. We agree that the results of our study have important implications for operational glaciogenic cloud seeding. It is important to note that our study focused solely on the size growth rates of ice crystals, whereas mass growth rates are more directly relevant for precipitation formation. Nevertheless, our results show that ice crystal growth rates increase continuously with increasing liquid water content (LWC), reinforcing the expectation that targeting clouds with high LWC is favorable. Furthermore, since our measurements focus on size growth rates, we observe peak growth around -5 °C. However, this could be misleading for precipitation goals, as mass growth of ice crystals peaks near -14 °C, where supersaturation with respect to ice and molecular diffusivity are jointly maximized. Therefore, we further suggest targeting clouds in this temperature range for the most efficient conversion of water into the ice phase. Finally, we observe that high ICNC lead to competition for the available water vapor, which limits both size and inherently also mass growth. For quick precipitation formation, larger ice crystals (in size and mass) are advantageous for two key reasons: (i) their larger surface area and greater terminal fall velocity promote riming, a dominant process for forming precipitation-sized hydrometeors; and (ii) larger hydrometeors are more likely to reach the ground without sublimating in potentially subsaturated layers below the cloud. Thus, our findings would imply that seeding strategies aiming to enhance precipitation should use lower seeding concentrations or distribute the seeding material over larger areas to avoid growth competition among ice crystals. This approach would also be beneficial from an environmental standpoint.

An open question that we are also curious about is whether it is more advantageous to seed at the cloud base, cloud center, or cloud top. This will be addressed in our follow-up project, which will continue to use cloud seeding techniques with a higher emphasis on precipitation formation.

We added a sentence in lines 51-57: *"In addition to providing fundamental insights into vapor diffusional ice crystal growth, our results offer valuable implications for cloud seeding operations, with the potential to improve their overall efficiency."*

We added a new paragraph in lines 422-429: *"In addition to providing fundamental insights into vapor diffusional ice crystal growth, our results also have important implications for operational glaciogenic cloud seeding. It is important to distinguish between growth in mass and growth in size of ice crystals, with maximum mass growth expected at temperatures around -14 °C (Takahashi et al., 1991). Consequently, if the efficiency of glaciogenic cloud seeding is evaluated based on the conversion of liquid water into ice, seeding will be most effective near this temperature. To enhance precipitation, our findings suggest targeting clouds with high LWC (Haupt et al., 2018; Rauber et al., 2019). Lower seeding concentrations or broader dispersal of the seeding agent would help to minimize competition for water vapor among ice crystals, promoting the growth of larger hydrometeors that are less likely to sublimate before reaching the ground."*

**References**

Chen, J., Rösch, C., Rösch, M., Shilin, A., and Kanji, Z. A.: Critical Size of Silver Iodide Containing Glaciogenic Cloud Seeding Particles, Geophysical Research Letters, 51, e2023GL106 680, https://doi.org/10.1029/2023GL106680, e2023GL106680 2023GL106680, 2024.

Haupt, S. E., Rauber, R. M., Carmichael, B., Knievel, J. C., and Cogan, J. L.: 100 Years of Progress in Applied Meteorology. Part I: Basic Applications, Meteorological Monographs, 59, 22.1–22.33, https://doi.org/10.1175/AMSMONOGRAPHS-D-18-0004.1, 2018.

Korolev, A. and Isaac, G.: Relative Humidity in Liquid, Mixed-Phase, and Ice Clouds, Journal of The Atmospheric Sciences - J ATMOS SCI, 63, 2865–2880, https://doi.org/10.1175/JAS3784.1, 2006.

Miles, N. L., Verlinde, J., and Clothiaux, E. E.: Cloud Droplet Size Distributions in Low-Level Stratiform Clouds, Journal of the Atmospheric Sciences, 57, 295–311, https://doi.org/10.1175/1520-0469(2000)057<0295:CDSDIL>2.0.CO;2, 2000.

Miller, A. J., Ramelli, F., Fuchs, C., Omanovic, N., Spirig, R., Zhang, H., Lohmann, U., Kanji, Z. A., and Henneberger, J.: Two New Multirotor Uncrewed Aerial Vehicles (UAVs) for Glaciogenic Cloud Seeding and Aerosol Measurements within the CLOUDLAB Project, Atmospheric Measurement Techniques, 17, 601–625, https://doi.org/10.5194/amt-17-601-2024, 2024.

Miller, A. J., Fuchs, C., Ramelli, F., Zhang, H., Omanovic, N., Spirig, R., Marcolli, C., Kanji, Z. A., Lohmann, U., and Henneberger, J.: Quantified Ice-Nucleating Ability of AgI-containing Seeding Particles in Natural Clouds, Atmospheric Chemistry and Physics, 25, 5387–5407, https://doi.org/10.5194/acp-25-5387-2025, 2025.

Omanovic, N., Ferrachat, S., Fuchs, C., Ramelli, F., Henneberger, J., Miller, A. J., Spirig, R., Zhang, H., and Lohmann, U.: Chasing Ice Crystals: Interlinking Cloud Microphysics and Dynamics in Cloud Seeding Plumes With Lagrangian Trajectories, Journal of Advances in Modeling Earth Systems, 17, e2025MS005 016, https://doi.org/10.1029/2025MS005016, e2025MS005016 2025MS005016, 2025.

Pruppacher, H. R. and Klett, J. D.: Microphysics of Clouds and Precipitation, no. 18 in Atmospheric and Oceanographic Sciences Library, Springer Netherlands, Dordrecht, 1 edn., ISBN 978-0-7923-4211-3 978-0-306-48100-0, https://doi.org/10.1007/978-0-306-48100-0, 2010.

Rauber, R. M., Geerts, B., Xue, L., French, J., Friedrich, K., Rasmussen, R. M., Tessendorf, S. A., Blestrud, D. R., Kunkel, M. L., and Parkinson, S.: Wintertime Orographic Cloud Seeding—A Review, Journal of Applied Meteorology and Climatology, 58, 2117–2140, https://doi.org/10.1175/JAMC-D-18-0341.1, 2019.

Scherrer, S. C. and Appenzeller, C.: Fog and Low Stratus over the Swiss Plateau - a Climatological Study, International Journal of Climatology, 34, 678–686, https://doi.org/10.1002/joc.3714, 2014.

Schmale, J., Henning, S., Decesari, S., Henzing, B., Keskinen, H., Sellegri, K., Ovadnevaite, J., Pöhlker, M. L., Brito, J., Bougiatioti, A., Kristensson, A., Kalivitis, N., Stavroulas, I., Carbone, S., Jefferson, A., Park, M., Schlag, P., Iwamoto, Y., Aalto, P., Äijälä, M., Bukowiecki, N., Ehn, M., Frank, G., Fröhlich, R., Frumau, A., Herrmann, E., Herrmann, H., Holzinger, R., Kos, G., Kulmala, M., Mihalopoulos, N., Nenes, A., O'Dowd, C., Petäjä, T., Picard, D., Pöhlker, C., Pöschl, U., Poulain, L., Prévôt, A. S. H., Swietlicki, E., Andreae, M. O., Artaxo, P., Wiedensohler, A., Ogren, J., Matsuki, A., Yum, S. S., Stratmann, F., Baltensperger, U., and Gysel, M.: Long-Term Cloud Condensation Nuclei Number Concentration, Particle Number Size Distribution and Chemical Composition Measurements at Regionally Representative Observatories, Atmospheric Chemistry and Physics, 18, 2853–2881, https://doi.org/10.5194/acp-18-2853-2018, 2018.

Takahashi, T., Endoh, T., Wakahama, G., and Fukuta, N.: Vapor Diffusional Growth of Free-Falling Snow Crystals between -3 and -23°C, Journal of the Meteorological Society of Japan. Ser. II, 69, 15–30, https://doi.org/10.2151/jmsj1965.69.1_15, 1991.

Wang, P. K. and Ji, W.: Collision Efficiencies of Ice Crystals at Low–Intermediate Reynolds Numbers Colliding with Supercooled Cloud Droplets: A Numerical Study, Journal of the Atmospheric Sciences, 57, 1001–1009, https://doi.org/10.1175/1520-0469(2000)057<1001:CEOICA>2.0.CO;2, 2000.

Westerhuis, S., Fuhrer, O., Cermak, J., and Eugster, W.: Identifying the Key Challenges for Fog and Low Stratus Forecasting in Complex Terrain, Quarterly Journal of the Royal Meteorological Society, 146, 3347–3367, https://doi.org/10.1002/qj.3849, 2020.